# DEEP NEURAL NETWORK INITIALIZATION WITH SPARSITY INDUCING ACTIVATIONS

**Ilan Price**[*,†]**, Nicholas Daultry Ball**[*]**, Samuel C.H. Lam**[*]**, Adam C. Jones**[*] **& Jared Tanner**[*]

(*) Mathematical Institute, University of Oxford
(†) The Alan Turing Institute
{ilan.price,nicholas.daultryball,samuel.lam,adam.c.jones,tanner}
@maths.ox.ac.uk

## ABSTRACT

Inducing and leveraging sparse activations during training and inference is a promising avenue for improving the computational efficiency of deep networks, which is increasingly important as network sizes continue to grow and their application becomes more widespread. Here we use the large width Gaussian process limit to analyze the behaviour, at random initialization, of nonlinear activations that induce sparsity in the hidden outputs. A previously unreported form of training instability is proven for arguably two of the most natural candidates for hidden layer sparsification; those being a shifted ReLU ($\phi(x) = \max(0, x - \tau)$ for $\tau \geq 0$) and soft thresholding ($\phi(x) = 0$ for $|x| \leq \tau$ and $x - \text{sign}(x)\tau$ for $|x| > \tau$). We show that this instability is overcome by clipping the nonlinear activation magnitude, at a level prescribed by the shape of the associated Gaussian process variance map. Numerical experiments verify the theory and show that the proposed magnitude clipped sparsifying activations can be trained with training and test fractional sparsity as high as 85% while retaining close to full accuracy.

## 1 INTRODUCTION

Improving the computational efficiency of large deep neural networks is the subject of many lines of work, including network weight pruning and quantization (Blalock et al., 2020), adapting network architectures (Bizopoulos & Koutsouris, 2021), and inducing low-dimensional structure in hidden layer outputs through low-rank, sparse, or similar structures (see Sec. 1.2). Two of the key quantities that determine the computational efficiency of the forward pass of a deep network are the total number of floating point operations (FLOPs) involved and the total amount of memory required. These quantities are primarily influenced by the tensor products of layer weights and layer inputs. The existence of efficient techniques for storing and computing with sparse vectors and matrices makes the sparsification of DNN weights and layer inputs/outputs a promising strategy for making the forward passes of deep networks more efficient. In contrast to the extensive literature on weight pruning and quantization, this paper investigates the training of networks with nonlinear activation functions that induce highly sparse hidden layer outputs throughout both training and inference. Sparse hidden layers are especially appealing as they naturally combine with the other methods by subselecting only a portion of the weight matrix active for that input.

To induce a target sparsity level we will leverage the fact that deep networks randomly initialized with i.i.d. entries generate hidden layers whose entries approach Gaussian distributions with known variance as the network width increases. For a survey of the Gaussian process behaviour of deep networks see Roberts et al. (2022) and references therein, and for initial derivations of feedforward networks (Lee et al., 2018), CNNs (Xiao et al., 2018), LSTMs and GRUs (Gilboa et al., 2019), RNNs (Chen et al., 2018), ResNets (Yang & Schoenholz, 2017) and extra features like dropout (Schoenholz et al., 2017; Huang et al., 2020) or batch normalization (Yang et al., 2019) and pruning (Hayou et al., 2021). This manuscript considers random initializations of both feedforward networks, abridged here as DNNs, of width $N_l$,

$$h_j^l = \sum_{i=1}^{N_l} W_{ij}^l x_i^{l-1} + b_j^l, \qquad x^l = \phi(h^l) \qquad \text{for } l = 1, 2, \ldots, L, \tag{1}$$

and one-dimensional CNNs[1] with input channel length $C_{in}^l$ and kernel width $2k + 1$,

$$h_j^l(\alpha) = \sum_{i=1}^{C_{in}^l} \sum_{\beta=-k}^{k} W_{ij}^l(\beta) x_i^{l-1}(\alpha + \beta) + b_j^l, \qquad x^l = \phi(h^l) \qquad \text{for } l = 1, 2, \ldots, L; \quad (2)$$

specifically, with a focus on activation functions $\phi$ that induce a prescribed substantial fractional sparsity at initialization, which in practice is maintained throughout training and inference on unseen data. To the best of our knowledge, highly sparsifying activations have not been studied at initialization, which may well be due to — as we will show — the failure of DNNs and CNNs to train with the intuitive choices of sparsifying activation functions. We analytically identify the source of this failure, which is due to an instability in the variance map of successive layers' Gaussian processes. That is, for arguably the most natural sparsifying activations, the Edge-of-Chaos (EoC) initialization Poole et al. (2016) needed to train very deep networks happens to coincide with introducing an instability that typically results in exponential growth of the Gaussian process variance with depth. Here, we introduce modified activation functions, using simple magnitude clipping, which decouples EoC initialization from the Gaussian process variance, and demonstrate that this allows training of very deep networks with activation sparsity up to 85%.

## 1.1 MAIN RESULTS

The Gaussian process model of DNNs (1) in the large width limit and CNNs (2) in the large channel limit are characterized by their associated variance. For DNNs of width $N_l$ with Gaussian weights $w_{i,j}$ drawn i.i.d. $\mathcal{N}(0, \sigma_w^2/N_l)$ and bias $b_j$ drawn i.i.d. $\mathcal{N}(0, \sigma_b^2)$ it was shown by Lee et al. (2018) that the pre-activation outputs $h_j^l$ approach $\mathcal{N}(0, q^l)$ where $q^l$ can be computed through the 'variance map' or 'length map'

$$q^l = V_\phi(q^{l-1}) := \sigma_w^2 \int_{\mathbb{R}} \left( \phi(\sqrt{q^{l-1}} \, z) \right)^2 \gamma(dz) + \sigma_b^2, \tag{3}$$

where $\gamma(dz)$ is the standard Gaussian measure $e^{-z^2/2}/\sqrt{2\pi} \, dz$, and $q^0 = \|x^0\|_2^2/N^0$, $q^1 = \sigma_w^2 q^0 + \sigma_b^2$. Note that the variance map $V_\phi(\cdot)$ depends on $\phi$, $\sigma_w^2$, and $\sigma_b^2$. Typically $V_\phi$ has a single nonzero fixed point, which is essential for the resulting theory[2].

The Gaussian process model for CNNs (2) in the large channel limit is somewhat more complex due to correlation within $h_j^l(\alpha)$ and $h_j^l(\alpha')$ when $|\alpha - \alpha'| < 2k + 1$. Xiao et al. (2018) proved that this Gaussian process is characterized by the matrix

$$\Sigma_{\alpha,\alpha'}^l = \sigma_b^2 + \frac{\sigma_w^2}{2k+1} \sum_{\beta=-k}^{k} \mathbb{E} \left( \phi(h_j^{l-1}(\alpha + \beta)) \phi(h_j^{l-1}(\alpha' + \beta)) \right), \tag{4}$$

where expectation is taken over the weights and biases. The full hidden layer vector $h^l$ then approaches the Gaussian distribution $\mathcal{N}(0, \mathcal{A} \star \sigma_{\alpha,\alpha'}^l)$ where $\mathcal{A} = \frac{1}{2k+1} I_{2k+1}$ and $\star$ denotes a two-dimensional circulant cross-correlation that accounts for the $2k + 1$ overlapping locations. The covariance matrix for the CNN Gaussian process follows a similar variance map as (3),

$$\Sigma_{\alpha,\alpha'}^l = q^l (\delta_{\alpha,\alpha'} + (1 - \delta_{\alpha,\alpha'} \rho^l)), \tag{5}$$

where $\rho^{l-1} = q_{\alpha,\alpha'}^{l-1}/\sqrt{q_\alpha^{l-1} q_{\alpha'}^{l-1}}$ is the correlation coefficient of the inputs at layer $l - 1$ (Poole et al., 2016). Assuming that both $q_a^l$ and $q_b^l$ converge quickly to $q^*$ yields the following iterative correlation map

$$\rho^l = R_\phi(\rho^{l-1}) = \frac{1}{q^*} \left( \sigma_w^2 \iint \phi(u_1) \phi(u_2) \gamma(dz_1) \gamma(dz_2) + \sigma_b^2 \right), \tag{6}$$

---

[1] Higher dimensional CNNs follow similarly but with more complex notation, see Lee et al. (2018).

[2] There are, however, some notable counter-examples, with ReLU having $V_{\text{ReLU}}(q) = q + \sigma_b^2$ and consequently when $\sigma_b = 0$, all $q$ are fixed points, while when $\sigma_b > 0$ the variance $q^l$ increases by $\sigma_b^2$. Conversely ELU with $\sigma_b = 0$ has $V_{\text{ELU}}(q) < q$ for all $q$ with $q^* = 0$ the only fixed point, see Murray et al. (2022).

where $u_1 = \sqrt{q^*} z_1$ and $u_2 = \sqrt{q^*}(\rho^{l-1}z_1 + \sqrt{1-(\rho^{l-1})^2}\, z_2)$, which describes how the correlation between two distinct inputs (in the DNN case) or two entries at different locations with overlapping kernels (in the CNN case) evolves from one layer to the next, after both starting with or converging to variance $q^*$.

Equipped with an accurate model of the hidden layer entries it is straightforward to compute the fractional sparsity that a nonlinear activation will induce, and moreover to design activation functions $\phi(\cdot)$ which induce a prescribed sparsity level in the activations. Perhaps the most obvious candidate for such sparsifying activation functions is a shifted ReLU, denoted here as $\mathsf{ReLU}_\tau$, and defined as

$$\mathsf{ReLU}_\tau(x) = \begin{cases} x - \tau, & \text{if } x > \tau \\ 0, & \text{otherwise,} \end{cases} \qquad (7)$$

where $\tau > 0$ allows greater sparsity than the standard ReLU when $\tau = 0$.[3] The other especially natural sparsifying activation is the SoftThreshold function $\mathsf{ST}_\tau$ defined as

$$\mathsf{ST}_\tau(x) = \begin{cases} x - \text{sign}(x)\tau, & \text{if } |x| > \tau \\ 0, & \text{otherwise,} \end{cases} \qquad (8)$$

which is an odd-function variant of the $\mathsf{ReLU}_\tau$; see Figure 1. The appeal of $\mathsf{ST}_\tau$ to induce sparsity is that it is an optimal denoiser (Donoho, 1995) for fixed values with additive Gaussian noise and is widely used in the compressed sensing community (Foucart & Rauhut, 2013) to encourage sparsity.

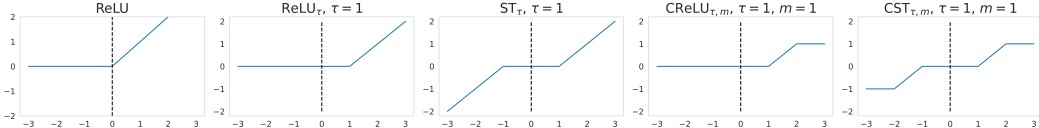

Figure 1: From left to right: ReLU (i.e. $\mathsf{ReLU}_\tau$ with $\tau = 0$), $\mathsf{ReLU}_\tau$ ($\tau = 1$), $\mathsf{ST}_\tau$ ($\tau = 1$), $\mathsf{CReLU}_{\tau,m}$ ($\tau = 1, m = 1$), $\mathsf{CST}_{\tau,m}$ ($\tau = 1, m = 1$)

Unfortunately, (7) and (8) both suffer from a previously unreported instability of their hidden layer variance map (3) which impedes training DNNs and CNNs with these activation functions. Specifically, both (7) and (8) have the property that when initialized on the EoC, their associated $V_\phi(q)$ has a single fixed point $q^*$ with $V'_\phi(q^*) = 1$, and moreover $V''_\phi(q^*) > 0$, causing $q^*$ to be only stable from the left, see Sec. A and B. Due to the natural stochasticity of $q^l$ for finite dimensional networks, even for moderate depths, $q^l$ typically obtains a value in excess of $q^*$ and then diverges exponentially. This phenomenon is unusual as most common activation functions have a unique stable fixed point with $V'_\phi(q^*) < 1$.

It is possible to regain the stability of the fixed point of the Gaussian process variance map by suitable modification of the sparsity-inducing activation functions $\phi$. This can be achieved by clipping the magnitude of these activations, (7) and (8); specifically, let

$$\mathsf{CReLU}_{\tau,m}(x) = \begin{cases} 0, & \text{if } x < \tau \\ x - \tau, & \text{if } \tau < x < \tau + m \\ m, & \text{if } x > \tau + m \end{cases} \qquad (9)$$

and

$$\mathsf{CST}_{\tau,m}(x) = \begin{cases} 0, & \text{if } |x| < \tau \\ x - \text{sign}(x)\tau, & \text{if } \tau < |x| < \tau + m \\ \text{sign}(x)m, & \text{if } |x| > \tau + m, \end{cases} \qquad (10)$$

as plotted in Figure 1. This simple modification is sufficient to guarantee $V'_\phi(q^*) < 1$ for $m$ bounded. Proof-of-concept experiments in Sec. 4 verify the efficacy of clipping to retain stability and show a trade-off with $m$ between the stability of training and the expressivity of the network. Moreover, these experiments show that the for DNNs full test accuracy of a standard ReLU network baseline can be retained, or even slightly improved, with activation sparsity as high as 85%. For CNNs a similar phenomenon is observed with full accuracy of ReLU baseline being retained for 70% sparsity, while 85% sparsity incurs just a 4% accuracy drop.

---

[3]This is a continuous variant of the Forced Activation Threshold ReLU considered in Kurtz et al. (2020).

## 1.2 RELATED WORK

As noted and referenced above, there is a large body of work analysing the infinite-width limit for stable initialization of very deep networks. Complementary to the sparse hidden layer results presented here is the extensive literature on weight pruning (sparsifying the weights), both after a network has been trained, as a type of network 'post-processing' (see e.g. Han et al. (2015)), and at initialization, such that the weights are sparse during training (Lee et al., 2019; Tanaka et al., 2020; Hayou et al., 2020). See Hoefler et al. (2021) for a review of the work on weight sparsification/pruning. In contrast, sparsification of the layer outputs—often referred to as the layer 'activations', or 'feature maps' in the case of CNNs—has received relatively little attention. Those few works which do focus on sparsity in the activations tend to propose methods to increase the sparsity of an existing, pre-trained network via fine-tuning. Georgiadis (2019) proposes inducing sparse activations (feature maps) in already-trained CNNs by fine-tuning them with added $L_1$ regularisation, penalising the $L_1$ norms of all feature maps. Kurtz et al. (2020) instead fine-tune ReLU networks with Hoyer regularisation, and introduce the FATReLU (defined as 0 if $x < \tau$ else $x$) for some threshold $\tau$ to increase the hidden layer sparsity.

## 1.3 ORGANISATION OF THE PAPER

Section 2 analyses natural candidates for sparsifying activation functions (7) and (8), to show why very deep networks using these activation functions will almost always fail to train for $\tau > 0$. Section 3, analyses the magnitude-clipped variants (9) and (10), and shows analytically that this circumvents the most important of the aforementioned inability of the networks to train. Section 4 validates the theory with proof-of-concept numerical experiments training DNNs and CNNs using the proposed activation functions. Finally, Section 5 suggests a number of lines of future work.

## 2 THE INSTABILITY OF $\mathsf{ReLU}_\tau$ AND $\mathsf{ST}_\tau$ AS SPARSIFYING ACTIVATION FUNCTIONS

The instability of $\mathsf{ReLU}_\tau$ and $\mathsf{ST}_\tau$ for $\tau > 0$ is a consequence of the shape of their associated variance maps $V_\phi(q)$, (3) when initialized on the EoC. EoC initialization requires that the slope of the correlation map $R_\phi(\rho)$ at $\rho = 1$

$$\chi_{1,\phi} := R_\phi'(1) = \sigma_w^2 \int_{\mathbb{R}} (\phi'(\sqrt{q^*}z))^2 \gamma(dz), \tag{11}$$

be equal to 1. $\chi_{1,\phi}$ determines the stability of the network to small perturbations of the inputs. When $\chi_{1,\phi} < 1$ the network maps perturbations together at an exponential rate (this is referred to as the ordered regime [4]), while when $\chi > 1$ the perturbations diverge at an exponential rate (the chaotic regime). Having $\chi_{1,\phi} = 1$ but $R(\rho) > \rho$ for $\rho$ near 1 results in a stable convergence of correlations at a sub-exponential rate, and also helps to prevent exploding or vanishing gradients; for details see Yang & Schoenholz (2017) and App. A. As noted in Section 1.1 preserving $\chi_{1,\phi} = 1$ depends on $q^l$ converging to $q^*$, and thus also requires that the variance map $V_\phi(q)$ be sufficiently stable around $q^*$. Unfortunately it is impossible for $\mathsf{ReLU}_\tau$ or $\mathsf{ST}_\tau$ to meet both of these requirements.

The variance maps $V_\phi(q)$ for $\phi = \mathsf{ReLU}_\tau$ and $\mathsf{ST}_\tau$ are computed in App. B, tabulated in Table 1, and plotted in Figure 2 with $(\sigma_w, \sigma_b)$ on the EoC for different values of $\tau$. The key observation is that for both $\mathsf{ReLU}_\tau$ and $\mathsf{ST}_\tau$, the slope $V_\phi'(q)$ at the fixed point $V_\phi(q^*) = q^*$ is equal to $\chi_{1,\phi}$, and thus we necessarily have that $V_\phi'(q^*) = 1$ on the EoC.

When $\tau = 0$, $\mathsf{ReLU}_\tau$ is just the standard ReLU function, and $\mathsf{ST}_\tau$ collapses to the identity. In both cases, when $(\sigma_w, \sigma_b)$ lie on the EoC, $V_{\mathsf{ReLU}_\tau}'(q) = 1$ for all $q$, making any $q$ a fixed point of $V_{\mathsf{ReLU}_\tau}$; see the left-most plot in Figure 2.

For $\tau > 0$ however, the second derivative $V_\phi''(q^*)$ is strictly positive, causing networks with $\mathsf{ReLU}_\tau$ and $\mathsf{ST}_\tau$ with $\tau > 0$ to have unstable Gaussian process variance propagation and consequently prove effectively impossible to train for large sparsity. We verify this claim experimentally – see App. C for the experimental details and Table 2 for the results.

---

[4]The ordered regime $\chi_{1,\phi} < 1$ is stable to perturbations, but for practical training is excessively so, with exponential convergence of all inputs to a single point.

| $\phi$ | $V'_\phi(q)$ | $\chi_{1,\phi}$ at $q = q^*$ | $V''_\phi(q)$ | $\hat{\tau}_\phi(s, q^*)$ | $\sigma_w^2$ |
|---|---|---|---|---|---|
| $\mathsf{ReLU}_\tau$ | $\sigma_w^2 \Phi\left(-\frac{\tau}{\sqrt{q}}\right)$ | $\sigma_w^2 \Phi\left(-\frac{\tau}{\sqrt{q^*}}\right)$ | $\Phi\left(-\frac{\tau}{\sqrt{q}}\right)^{-1} \frac{\tau e^{-\frac{\tau^2}{2q}}}{2\sqrt{2\pi}q^{\frac{3}{2}}}$ | $\sqrt{q^*}\Phi^{-1}(s)$ | $\frac{1}{1-s}$ |
| $\mathsf{ST}_\tau$ | $2\sigma_w^2 \Phi\left(-\frac{\tau}{\sqrt{q}}\right)$ | $2\sigma_w^2 \Phi\left(-\frac{\tau}{\sqrt{q^*}}\right)$ | $2\Phi\left(-\frac{\tau}{\sqrt{q}}\right)^{-1} \frac{\tau e^{-\frac{\tau^2}{2q}}}{2\sqrt{2\pi}q^{\frac{3}{2}}}$ | $\sqrt{2q^*}\mathrm{erf}^{-1}(s)$ | $\frac{1}{1-s}$ |

Table 1: Variance and correlation map derivatives for $\mathsf{ReLU}_\tau$ and $\mathsf{ST}_\tau$ where $\Phi(\cdot)$ is the CDF of a standard normal distribution. The correspondence of $V'_\phi$ and $\chi_{1,\phi}$ means that EoC initialization implies that $V'_\phi(q^*) = 1$ for all $\tau > 0$. Moreover, the second derivative of the variance map $V''_\phi$ is strictly positive for every $q$, resulting in instability of the fixed point $V_\phi(q^*) = q^*$. The simplified expression of $\sigma_w^2$ in the final column occurs when $\chi_{1,\phi} = 1$.

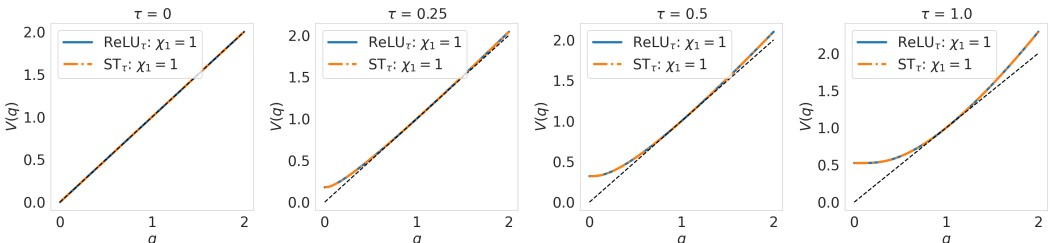

Figure 2: Variance maps for $\mathsf{ReLU}_\tau$ and $\mathsf{ST}_\tau$ with $(\sigma_w, \sigma_b)$ on the EoC, for different values of $\tau$. Here $q^* = 1$ is used to compute $\chi_{1,\mathsf{ReLU}_\tau}$. The dashed line is the identity map. The curves for $\mathsf{ReLU}_\tau$ and $\mathsf{ST}_\tau$ overlap exactly for a given $\tau$. Note, however, that a fixed value of $\tau$ corresponds to substantially different activation sparsities for $\mathsf{ReLU}_\tau$ and $\mathsf{ST}_\tau$. Figure 9 in App. C compares $V(q)$ for fixed output sparsities.

In practice, we choose $\tau$ to control the 'expected sparsity' $s$ (proportion of zeros) after applying different activation functions. Assuming the activation function is applied to a random Gaussian vector with independent entries, each with variance $q^*$, we have

$$s = \mathbb{P}(\phi(Z) = 0), \quad Z \sim \mathcal{N}(0, q^*). \tag{12}$$

The optimal (smallest) $\tau$ for a DNN with activation function $\phi = \mathsf{ReLU}_\tau$ or $\mathsf{ST}_\tau$ and activation sparsity $s$, denoted as $\hat{\tau}_\phi(s, q^*)$, is computed in App. F and tabulated in Table 1. Substituting $\hat{\tau}_\phi(s, q^*)$ into the expression for $\chi_{1,\phi}$ in Table 1 results in the value of $\sigma_w^2$ on the EoC, which turns out to be independent of $q^*$ for both $\mathsf{ReLU}_\tau$ and $\mathsf{ST}_\tau$, and which thus makes achieving approximate dynamical isometry (Pennington et al., 2018) with these networks impossible, see App A.

## 3 GAINING STABILITY AND TRAINABILITY VIA CLIPPING

Clipping the maximum output magnitude of $\mathsf{ReLU}_\tau$ and $\mathsf{ST}_\tau$ allows the EoC condition $\chi_{1,\phi} = 1$ to be satisfied while ensuring $V'_\phi(q^*) < 1$. The clipped activation functions are denoted as $\mathsf{CReLU}_{\tau,m}$ and $\mathsf{CST}_{\tau,m}$ in (9) and (10) respectively. This allows $\mathsf{CReLU}_{\tau,m}$ and $\mathsf{CST}_{\tau,m}$ to generate hidden layer outputs which are both very sparse and trainable when initialized at the EoC.

The associated variance and correlation maps for $\mathsf{CReLU}_{\tau,m}$ and $\mathsf{CST}_{\tau,m}$ are computed and plotted for a variety of sparsity factors $\tau$ and clipping levels $m$ in App. E. The stability of the fixed point $q^*$ when $\chi_{1,\phi} = 1$ follows from the expressions for the gradient of their variance map $V'_\phi(q)$ and the slope $\chi_{1,\phi}$:

$$V'_{\mathsf{CReLU}_{\tau,m}}(q) = \sigma_w^2 \left(-\frac{m}{\sqrt{2\pi q}}e^{-\frac{1}{2}\left(\frac{m+\tau}{\sqrt{q}}\right)^2} + \frac{1}{2}\left(\mathrm{erf}\left(\frac{m+\tau}{\sqrt{2q}}\right) - \mathrm{erf}\left(\frac{\tau}{\sqrt{2q}}\right)\right)\right), \tag{13}$$

$$V'_{\mathsf{CST}_{\tau,m}}(q) = 2V'_{\mathsf{CST}_{\tau,m}}(q), \tag{14}$$

$$\chi_{1,\mathsf{CReLU}_{\tau,m}} = \frac{\sigma_w^2}{2}\left(\mathrm{erf}\left(\frac{m+\tau}{\sqrt{2q^*}}\right) - \mathrm{erf}\left(\frac{\tau}{\sqrt{2q^*}}\right)\right), \tag{15}$$

$$\chi_{1,\mathsf{CST}_{\tau,m}} = 2\chi_{1,\mathsf{CReLU}_{\tau,m}}. \tag{16}$$

Crucially,

$$\chi_{1,\mathsf{CReLU}_{\tau,m}} = V'_{\mathsf{CReLU}_{\tau,m}}(q^*) + \frac{\sigma_w^2 m}{\sqrt{2\pi q^*}} \exp\left(-\frac{(m+\tau)^2}{2q^*}\right) > V'_{\mathsf{CReLU}_{\tau,m}}(q^*), \tag{17}$$

$$\chi_{1,\mathsf{CST}_{\tau,m}} = V'_{\mathsf{CST}_{\tau,m}}(q^*) + \frac{\sqrt{2}\,\sigma_w^2 m}{\sqrt{\pi q^*}} \exp\left(-\frac{(m+\tau)^2}{2q^*}\right) > V'_{\mathsf{CST}_{\tau,m}}(q^*). \tag{18}$$

Thus, $V'_\phi(q^*) < 1$ at the EoC ($\chi_\phi = 1$), making $q^*$ *locally* stable for both $\phi = \mathsf{CReLU}_{\tau,m}$ or $\mathsf{CST}_{\tau,m}$. The local convergence rate of $q^l$ to the fixed point $q^*$ is determined by $1 - V'_\phi(q^*)$. Note, however, that certain values of $m$ and $s$ induce multiple fixed points, causing $q^*$ not to be *globally* stable; see Figure 4b as well as the bottom right panels of Figures 10 and 11 in App E. Another effect of bounding these activation functions is that $\sigma_w^2$ at the EoC ($\chi_{1,\phi} = 1$) is no longer independent of $q^*$ for a given expected sparsity $s$, although this unfortunately still does not enable us to achieve approximate dynamical isometry for these activation functions, for details see App. F.

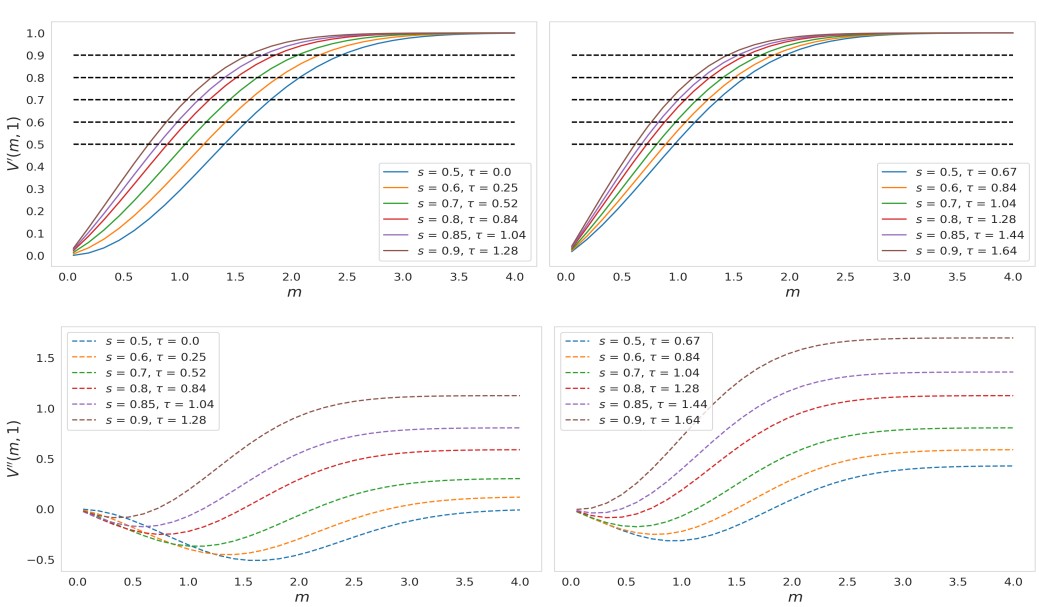

Figure 3: Plots of $V'_\phi(m)$ (upper) and $V''_\phi(m)$ (lower) and at $q^* = 1$ for $\mathsf{CReLU}_{\tau,m}$ (left) and $\mathsf{CST}_{\tau,m}$ (right) with for different fractional sparsities. For $V'_\phi(m)$ the horizontal dashed black lines are plotted at 0.5, 0.6, 0.7, 0.8, 0.9.

## 3.1 CHOOSING THE CLIPPING MAGNITUDE $m$ FOR $\mathsf{CReLU}_{\tau,m}$ AND $\mathsf{CST}_{\tau,m}$.

A larger clipping hyperparameter $m$ increases the expressivity of the network by allowing larger activation values and a wider trainable (non-zero gradient) region of its domain. It also plays a role in determining the shape of the variance map—crucially, $V'_\phi(q^*)$ (see Equation (13)), as well as the curvature $V''_\phi(q^*)$. Figure 3 show $V'_\phi(q^*)$ and $V''_\phi(q^*)$ as functions of $m$ for different sparsity levels. As $m$ increases, $V'_\phi(q^*)$ increases and tends to 1 from below as $m \to \infty$.

While $V'_\phi(q^*) < 1$ for each $m$, the curvature $V''_\phi(q^*)$ initially decreases to negative values and then increases as $m$ increases from 0. Stability for finite-dimensional networks requires $V_\phi(q)$ to be sufficiently stable around $q^*$ so that the natural stochasticity in $q^l$ remains in the local stable region of $V_\phi(q)$, meaning that $q^l$ does in practice remain approximately equal to $q^*$. In particular, for larger values of $m$ the curvature $V''_\phi(q^*)$ can cause $V_\phi(q)$ curve up after crossing the $V_\phi(q) = q$ line and approximately follow it for an extended interval before flattening out again—resulting in

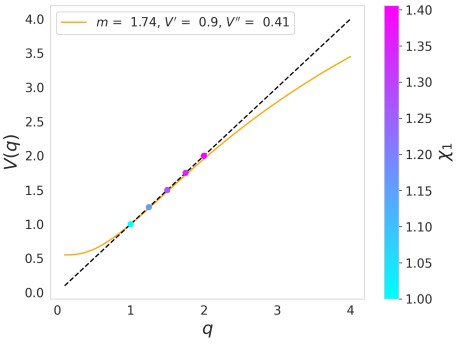 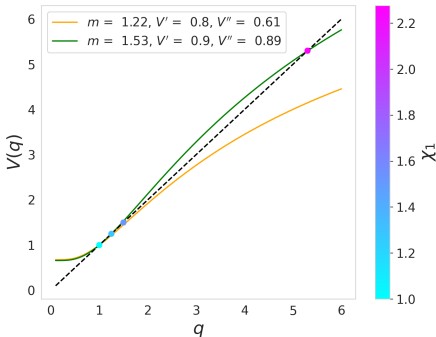

(a) $\phi = \mathsf{CReLU}_{\tau,m}$, $s = 0.85$, $m$ chosen such that $V'_\phi(q^*) = 0.9$ (orange), with $q^* = 1$.

(b) $\phi = \mathsf{CST}_{\tau,m}$, $s = 0.85$, $m$ chosen such that $V'_\phi(q^*) = 0.8$ (orange) and 0.9 (green), with $q^* = 1$.

Figure 4: Variance maps for $\mathsf{CReLU}_{\tau,m}$ in (a) and $\mathsf{CST}_{\tau,m}$ in (b) at 85% sparsity and $m$ chosen such that even though $V'_\phi(q^*) < 1$ the network is unstable to train. For $\mathsf{CReLU}_{\tau,m}$ training fails due to $V_\phi(q)$ being approximately $q$ for a sufficiently large region around $q^*$ that the effective value of $\chi_1$ varies above 1 which results in exploding gradients. For $\mathsf{CST}_{\tau,m}$ training fails for the same reason of effective $\chi_1 > 1$, which occurs even more dramatically as $V_\phi(q)$ exceeds $q$ and has a second, unstable, fixed point as well as another stable fixed point, but with $\chi_1$ substantially greater than 1.

an extended region in which $|V_\phi(q) - q|$ is small. Increasing $m$ (and thus $V''\phi(q^*)$) even further causes a bifurcation, with two new fixed points emerging: a locally unstable one close to the original $q^*$, and another locally stable one at a larger value of $q$. These new fixed points cause a failure of $q^l$ to converge to $q^*$ in practice – instead, $q^l$ converges to or hovers at a value greater than $q^*$, corresponding to $\chi_{1,\phi} > 1$, which results in exploding gradients. See Figure 4b for illustrations of these phenomena which correspond to examples in Sec. 4 where networks fail to train. For this reason, the values of $m$ that have the best values of accuracy and ability to train are as large as possible while $V'_\phi(q^*)$ and $V''_\phi(q^*)$ remain sufficiently small to ensure practical stability of $q^*$ based on stochasticity of $q^l$ in finite-width networks; see Table 2 for examples of parameters and associated test accuracy for $\mathsf{CReLU}_{\tau,m}$ and $\mathsf{ST}_\tau$ respectively.

## 4 EXPERIMENTS

To confirm the theoretical predictions, we train both feedforward networks (abridged as DNNs) of width 300 and depth 100 using $\mathsf{CReLU}_{\tau,m}$ and $\mathsf{CST}_{\tau,m}$ to classify digits from the MNIST dataset and, similarly, CNNs with 300 channels in each layer and depth 50 are trained to classify images from the CIFAR10 dataset. Training such deep DNNs and CNNs allows us to verify the theory, which aims to improve information propagation at initialization through many layers. The absolute accuracy of the networks is not the focus of these experiments, rather it is the ability to retain trainability and approximately the accuracy of standard ReLU networks for sparsities greater than 50%. The networks are initialized at the EoC using $q^* = 1$, before being trained by stochastic gradient descent (SGD) for 200 epochs with learning rate of $10^{-4}$ and $10^{-3}$ for the DNN and CNN respectively. Each experiment is conducted with sparsity $s = 0.6, 0.7, 0.8,$ and $0.85$ and $m$ selected to have $V'_\phi(q^*) = 0.5, 0.7,$ and $0.9$. The results are shown in Table 2. These results should be contrasted with those for $\mathsf{ReLU}_\tau$ and $\mathsf{ST}_\tau$, also shown in Table 2, and described in App. C.

Whereas EoC-initialized $\mathsf{ReLU}_\tau$ and $\mathsf{ST}_\tau$ DNNs failed to train consistently even when initialized at sparsity levels 60% and 50% respectively, the EoC-initialized $\mathsf{CReLU}_{\tau,m}$ and $\mathsf{CST}_{\tau,m}$ DNNs can maintain full accuracy when initialized with up to 85% sparsity. $\mathsf{CReLU}_{\tau,m}$ and $\mathsf{CST}_{\tau,m}$ CNNs retain full accuracy for sparsity as high as 70% while 85% sparsity incurs a modest loss of accuracy to 66%. Conversely, their non-clipped counterparts fail to train at all at 60% and 70% sparsity respectively. Moreover, in those cases where training is stable and high accuracy is achieved, the starting sparsity level is maintained throughout training, resulting in trained models which exhibit approximately the same high activation sparsity levels when measured on the test set.

| | $s$ | $\tau$ | $m$ | $V'_\phi(q^*)$ | $V''_\phi(q^*)$ | DNN on MNIST Test accuracy mean | std | Test sparsity mean | std | CNN on CIFAR10 Test accuracy | Test sparsity |
|---|---|---|---|---|---|---|---|---|---|---|---|
| ReLU$_\tau$ | 0.50 | 0.00 | N/A | 1.0 | 0.0 | 0.94 | 0.002 | 0.50 | 0.001 | 0.70 | 0.52 |
| | 0.60 | 0.25 | N/A | 1.0 | 0.12 | 0.76 | 0.37 | 0.49 | 0.27 | 0.68 | 0.6 |
| | 0.70 | 0.52 | N/A | 1.0 | 0.3 | 0.10 | 0.00 | 0.00 | 0.00 | 0.1 | 0.0 |
| ST$_\tau$ | 0.5 | 0.67 | N/A | 1.0 | 0.43 | 0.10 | 0.00 | 0.00 | 0.00 | 0.1 | 0.0 |
| | 0.6 | 0.84 | N/A | 1.0 | 0.59 | 0.10 | 0.00 | 0.00 | 0.00 | 0.1 | 0.0 |
| | 0.7 | 1.04 | N/A | 1.0 | 0.81 | 0.10 | 0.00 | 0.00 | 0.00 | 0.1 | 0.0 |
| CReLU$_{\tau,m}$ | 0.60 | 0.25 | 1.22 | 0.5 | -0.44 | 0.92 | 0.004 | 0.60 | 0.001 | 0.70 | 0.61 |
| | | | 1.63 | 0.7 | -0.42 | 0.92 | 0.003 | 0.60 | 0.004 | 0.69 | 0.61 |
| | | | 2.25 | 0.9 | -0.19 | 0.92 | 0.01 | 0.60 | 0.01 | 0.69 | 0.60 |
| | 0.70 | 0.52 | 1.05 | 0.5 | -0.37 | 0.93 | 0.01 | 0.70 | 0.002 | 0.70 | 0.70 |
| | | | 1.45 | 0.7 | -0.31 | 0.92 | 0.003 | 0.70 | 0.002 | 0.69 | 0.69 |
| | | | 2.05 | 0.9 | -0.04 | 0.92 | 0.01 | 0.70 | 0.01 | 0.68 | 0.69 |
| | 0.80 | 0.84 | 0.89 | 0.5 | -0.24 | 0.94 | 0.004 | 0.80 | 0.002 | 0.64 | 0.80 |
| | | | 1.27 | 0.7 | -0.12 | 0.93 | 0.01 | 0.80 | 0.01 | 0.64 | 0.78 |
| | | | 1.85 | 0.9 | 0.21 | 0.94 | 0.01 | 0.79 | 0.02 | 0.65 | 0.78 |
| | 0.85 | 1.04 | 0.81 | 0.5 | -0.14 | 0.78 | 0.16 | 0.85 | 0.004 | 0.65 | 0.85 |
| | | | 1.17 | 0.7 | 0.02 | 0.94 | 0.004 | 0.85 | 0.003 | 0.65 | 0.84 |
| | | | 1.74 | 0.9 | 0.41 | 0.28 | 0.37 | 0.79 | 0.04 | 0.66 | 0.83 |
| CST$_{\tau,m}$ | 0.50 | 0.67 | 0.97 | 0.5 | -0.32 | 0.92 | 0.004 | 0.51 | 0.01 | 0.69 | 0.49 |
| | | | 1.36 | 0.7 | -0.23 | 0.92 | 0.01 | 0.50 | 0.004 | 0.69 | 0.48 |
| | | | 1.96 | 0.9 | 0.08 | 0.92 | 0.004 | 0.49 | 0.01 | 0.71 | 0.49 |
| | 0.60 | 0.84 | 0.89 | 0.5 | -0.24 | 0.93 | 0.01 | 0.60 | 0.003 | 0.68 | 0.59 |
| | | | 1.27 | 0.7 | -0.12 | 0.92 | 0.004 | 0.60 | 0.003 | 0.67 | 0.59 |
| | | | 1.85 | 0.9 | 0.21 | 0.93 | 0.004 | 0.59 | 0.01 | 0.67 | 0.57 |
| | 0.70 | 1.04 | 0.81 | 0.5 | -0.14 | 0.94 | 0.005 | 0.70 | 0.004 | 0.68 | 0.69 |
| | | | 1.17 | 0.7 | 0.02 | 0.93 | 0.005 | 0.71 | 0.01 | 0.68 | 0.67 |
| | | | 1.74 | 0.9 | 0.41 | 0.27 | 0.36 | 0.41 | 0.18 | 0.68 | 0.68 |
| | 0.80 | 1.28 | 0.72 | 0.5 | 0.00 | 0.92 | 0.04 | 0.80 | 0.001 | 0.66 | 0.79 |
| | | | 1.06 | 0.7 | 0.23 | 0.95 | 0.01 | 0.80 | 0.01 | 0.65 | 0.78 |
| | | | 1.61 | 0.9 | 0.69 | 0.11 | 0.01 | 0.15 | 0.01 | 0.31 | 0.18 |
| | 0.85 | 1.44 | 0.67 | 0.5 | 0.11 | 0.76 | 0.16 | 0.85 | 0.005 | 0.63 | 0.84 |
| | | | 1.00 | 0.7 | 0.39 | 0.93 | 0.01 | 0.85 | 0.01 | 0.63 | 0.83 |
| | | | 1.53 | 0.9 | 0.89 | 0.14 | 0.04 | 0.11 | 0.01 | 0.29 | 0.12 |

Table 2: Experimental results for ReLU$_\tau$ and ST$_\tau$ with sparsity $s$ up to $0.7$, and for CReLU$_{\tau,m}$ and CST$_{\tau,m}$ with sparsity $s$ up to $0.85$ and $m$ selected to have $V'_\phi(q^*) = 0.5, 0.7$, and $0.9$. DNNs have 100 layers and are tested on MNIST with 5 runs and the mean and standard deviation are reported for both test accuracy and average activation sparsity calculated on the test set. For the 50-layer CNN experiments on CIFAR10 a single experiment was conducted for each hyperparameter combination.

Upon analysing the 'failure' cases for DNN in which low accuracy is achieved, two failure modes are identified. The first of these corresponds to the analysis and predictions made in Section 3.1. For CReLU$_{\tau,m}$ and $s = 0.6, 0.7$, and $0.8$, we see negative or small positive values of $V''_\phi(q^*)$ for all values of $V'_\phi(q^*)$; this corresponds to a variance map at initialization with sufficient symmetry around $q^*$ to ensure that $q^l$ stays, in practice, very close to $q^*$ on average, thus avoiding any exploding gradients, and preserving trainability. Once $s$ increases to $0.85$, however, we start to observe the potential for $m$ to be too large—when $m$ is set such that $V'_\phi(q^*) = 0.9$, then $V''_\phi(q^*) = 0.41$, a high chance of failure in training is observed for the reasons described in Section 3.1 and highlighted by the variance map in Figure 4. Indeed, training fails in four out of five experiments, reflected by the low mean test accuracy and high standard deviation. Figure 5a plots the gradient norms at each layer for the first 15 steps of training for one of these failed runs. Even at the very first step, the gradients grow as the backwards pass proceeds (up to $O(10^3)$), and this reaches $O(10^6)$ by step 3[5].

The same trend is observed for DNNs and $s \geq 0.8$ with $\phi = $ CST$_{\tau,m}$. Starting from the lowest tested value of $m$ in these cases, increasing $m$ initially increases test accuracy, but at a certain point,

---

[5]The reason for the drop back to very low levels at later steps is due to the fact that once the weights (and thus activations) grow large enough, they almost all begin to land in the zero-gradient interval of the activation functions, resulting in no further gradient flow through those layers.

$m$ becomes too large, such that the corresponding combination of $V'_\phi(q^*)$ and $V''_\phi(q^*)$ cause failure in some (and then all) of the five experiments. Figure 5b shows the layerwise gradient norms over the first 15 training steps of a failed run with $\mathsf{CST}_{\tau,m}$ with $s = 0.85$ and $m = 1.22$. Though on step 1 the gradients remain $\sim O(1)$, very quickly the exploding gradient phenomenon emerges.

In contrast, Table 2 shows that CNNs with $\mathsf{CReLU}_{\tau,m}$ exhibit a more stable test accuracy, whereas the analogous tests for $\mathsf{CST}_{\tau,m}$ show a loss of accuracy for large $m$ where $V'_\phi(q^*) = 0.9$ once $V''_\phi(q^*) \geq 0.69$ at $s = 0.8$ and $m = 1.61$ as well as $s = 0.85$ and $m = 1.53$.

The second failure mode accounts for the failure to train to high accuracy once starting sparsity reaches 85% for low values of $m$ (the failure when $s = 0.85$ and $m$ is large is captured in the first failure mode); specifically, the failure of DNNs to obtain high accuracy when $s = 0.85$ and $m = 0.81$ for $\mathsf{CReLU}_{\tau,m}$, as well as when $s = 0.85$ and $m \leq 0.83$ for $\mathsf{CST}_{\tau,m}$. In all of these cases, EoC initialization avoids exploding gradients at initialization and performance does improve during training for all of the five runs. However, the resulting accuracy is reduced, reaching a maximum of $\approx 78\%$ when $s = 0.85$ at $m = 0.81$, for example. This is discussed further in App. G.

One final observation from Table 2 is that for DNNs the maximum accuracy actually increases with increasing sparsity (up to a point, and for sufficiently large values of $m$). Maximum accuracy for $\mathsf{ReLU}_\tau$ networks is achieved when sparsity is 80% and 85% (accuracy of 94%), and for $\mathsf{CST}_{\tau,m}$ networks when sparsity is 80% (accuracy of 95%). However, CNNs in contrast do not show the aforementioned improved accuracy as a result of sparsity, instead they exhibit a more gradual loss of accuracy and train for a somewhat larger range of $m$.

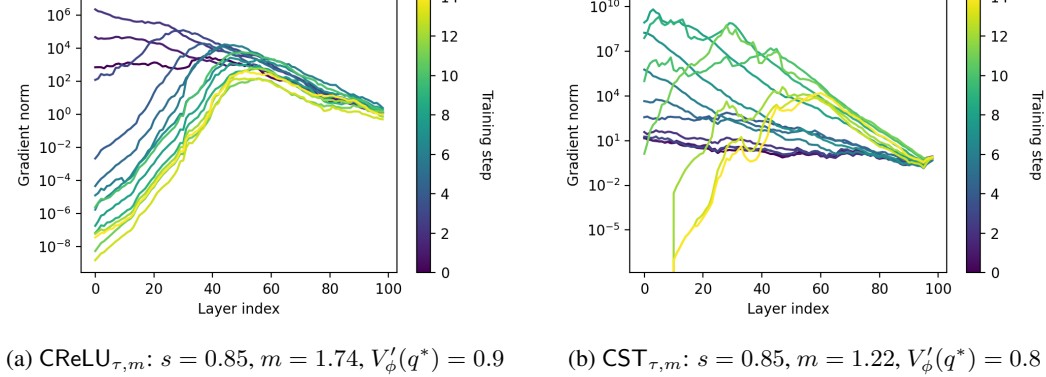

(a) $\mathsf{CReLU}_{\tau,m}$: $s = 0.85$, $m = 1.74$, $V'_\phi(q^*) = 0.9$    (b) $\mathsf{CST}_{\tau,m}$: $s = 0.85$, $m = 1.22$, $V'_\phi(q^*) = 0.8$

Figure 5: Gradient norms per layer for the first 15 steps of a failed training run of DNNs on MNIST.

## 5    CONCLUSIONS AND AVENUES FOR FUTURE WORK

The analysis here explains why arguably the most natural highly-sparsifying activation functions $\mathsf{ReLU}_\tau$ and $\mathsf{ST}_\tau$ fail to train for deep DNNs and CNNs. This instability is then overcome with a simple modification in the form of a bound on the absolute value of these functions, yielding a stable EoC initializations for the resulting networks, which allows for training very deep networks with these modified activation functions with activation sparsity up to 85%.

There are multiple natural and promising avenues to extend this line of work. Firstly, a more complete explanation of the unstable training dynamics when $s$ is very large and $m$ is small (the second failure mode identified in Section 4). Secondly, extension of EoC analysis for these sparsifying activation functions from DNNs and CNNs to transformer architectures, and networks with skip connections like ResNet would be important to have the greatest impact in recent applications. Additionally, it would be interesting to investigate whether smooth variants of $\mathsf{CReLU}_{\tau,m}$ and or $\mathsf{CST}_{\tau,m}$ exhibit any preferable properties over their piecewise-linear variants, specifically through controlling higher order moments of the spectrum of the input-output Jacobian. Finally, derivations of formulae for the largest choice of $m$ for which the variance map is stable to the variance of $q^l$ determined by the network width, input channels or other numbers of network parameters.

## REPRODUCIBILITY

Derivations of the theory in Section 2 and 3 can be found in Appendices B and E. Experimental setup and hyperparamters are detailed in Section 4 and Appendices C and D.

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

## A  GAUSSIAN PROCESS: VARIANCE, COVARIANCE, AND GRADIENT PROPAGATION AT INITIALIZATION

Numerous prior works Poole et al. (2016); Schoenholz et al. (2017); Pennington et al. (2017); Murray et al. (2022) have considered how the hyperparameters $\sigma_w^2$ and $\sigma_b^2$, and the choice of activation function $\phi$, affect signal and information propagation through the network at initialization, as well as the network's trainability via gradient backpropagation. The standard approach in these works is to perform a mean-field analysis in which one considers each layer to have infinite width, that is, $N^l \longrightarrow \infty$ for all $l$. In particular (Poole et al., 2016) considered the (width-scaled) length of the pre-activations.

The shape of both $V_\phi$ and $R_\phi$, along with the number of fixed points they admit and their associated stabilities depend on $\phi$, $\sigma_w^2$ and $\sigma_b^2$. Of particular interest is the slope of this correlation map at $\rho^* = 1$, which is given by

$$\chi_{1,\phi} := R_\phi'(1) = \sigma_w^2 \int_{\mathbb{R}} \left( \phi'(\sqrt{q^*}\, z) \right)^2 \gamma(dz). \tag{19}$$

When $\chi_{1,\phi} < 1$, the fixed point $\rho^*$ is stable, and any pair of points will asymptotically converge exponentially fast to being exactly aligned, whereas when $\chi_{1,\phi} > 1$, the fixed point $\rho^*$ is unstable, and another fixed point emerges $\rho^* < 1$. In the latter case, even two input points which are arbitrarily close to one another will asymptotically decorrelate. The former $\chi_{1,\phi} < 1$ regime is referred to in the literature as the 'ordered' regime, and the latter $\chi_{1,\phi} > 1$ regime is dubbed the 'chaos' regime. The 'Edge of Chaos' (EoC) is the set of points $(\sigma_w, \sigma_b)$ which separate these two regions, that is, the set of points $(\sigma_w, \sigma_b)$ such that $V(q^*) = q^*$ and $\chi_{1,\phi} = 1$.

It has been proven that initialising a network at the Edge of Chaos improves the depth of information propagation at initialization Yang & Schoenholz (2017). Moreover, as shown in Schoenholz et al. (2017), EoC initialization also helps to prevent exploding and vanishing gradients early in training, which otherwise make a network untrainable. To understand why this is the case, consider the backpropagation algorithm with which deep networks are trained via (stochastic) gradient-based optimisation methods. Given a loss function $\mathcal{L}(x, \theta)$ where $\theta = \bigcup_{l=1}^{L} \{W^l\} \cup \{b^l\}$, backpropagation calculates the gradients with respect to the weights and biases according to the follow recurrence relations

$$\frac{\partial \mathcal{L}(x, \theta)}{\partial h^L} := \delta^L = D^L \nabla_{h^l} \mathcal{L}(x, \theta), \tag{20}$$

$$\frac{\partial \mathcal{L}(x, \theta)}{\partial h^l} := \delta^l = (D^l W^{l+1})^\top \delta^{l+1}, \tag{21}$$

$$\frac{\partial \mathcal{L}(x, \theta)}{\partial b_i^l} = \delta_i^l, \tag{22}$$

$$\frac{\partial \mathcal{L}(x, \theta)}{\partial w_{i,j}^l} = h_j^{l-1} \delta_i^l, \tag{23}$$

where $D^l$ is the diagonal matrix with entries $D_{ii}^l = \phi'(h_i^l)$. In Pennington et al. (2018) it was shown that in the infinite-width limit, though error vectors $\delta^l$ are not Gaussian-like $h^l$, the second moment of the error vectors distribution evolves according to its own recurrence relation which depends on the value of $\chi_{1,\phi}$. Specifically, they show that (given the simplifying assumption that weights used during forward and backward passes are drawn independently from each other) $\tilde{q}^l := \mathbb{E}[(\delta_i^l)^2]$ evolves according to

$$\tilde{q}^l = \tilde{q}^{l+1} \frac{N_{l+1}}{N_l} \chi_{1,\phi}. \tag{24}$$

Thus when $\chi_{1,\phi} > 1$, the gradient norm is expected to increase exponentially over the course of the backward pass (a phenomenon known as 'exploding gradients'), and conversely to shrink exponentially (known as 'vanishing gradients') when $\chi_{1,\phi} < 1$. Both of these phenomena impede successful training of the network. In contrast, when $\chi_{1,\phi} = 1$ (and all layers have the equal width), gradient norm is preserved in expectation from one layer to the next, which allows for meaningful updates to be applied to all parameters.

The $\chi_{1,\phi} = 1$ condition also emerges from the analysis of the input-output Jacobian using tools from free probability in Pennington et al. (2017). The input-output Jacobian $J$ is given by

$$J = \frac{\partial x^L}{\partial x^0} = \prod_{l=1}^{L} D^l W^l. \tag{25}$$

The Jacobian is (as expected) closely related to the operations involved in backpropagation - and more precisely involves the product of the linear backpropagation operators applied to compute each layer's error vector during the backwards pass (see Equation (21)). In Pennington et al. (2018), the authors consider the limiting spectral density of $JJ^\top$ with moment generating function

$$M_{JJ^\top}(z) = \sum_{k=1}^{\infty} \frac{m_k}{z^k}. \tag{26}$$

To briefly recap the key points of the analysis in Pennington et al. (2018), assuming normalised input such that $q^1 = q^*$, then the distribution of $D^l$ is independent of $l$, and the moments of $D$ are given by

$$\mu_k = \int (\phi'(\sqrt{q^*}z))^{2k} \gamma(dz). \tag{27}$$

The first two of these ($\mu_1$ and $\mu_2$) define the first two moments of the spectrum of $JJ^\top$, $m_1$ and $m_2$, according to

$$m_1 = (\sigma_w^2 \mu_1)^L = (\chi_{1,\phi})^L \tag{28}$$

$$m_2 = (\sigma_w^2 \mu_1)^{2L} L \left( \frac{\mu_1}{\mu_2^2} + \frac{1}{L} - 1 - s_1 \right), \tag{29}$$

where $s_1$ is the first moment of the $S$-transform of $WW^\top$. When $W$ is Gaussian with mean 0 and variance $\sigma_w^2/N_{l-1}$ (as in the setup considered here), $s_1 = -1$ . In Pennington et al. (2018), the authors also consider the case when $W$ is orthogonal, in which case $s_1 = 0$.

From Equation (28), $m_1 = (\chi_{1,\phi})^L$, which implies that for large $L$, the scale of the error vectors in early layers is likely to blow up or collapse to 0 if $\chi_{1,\phi} > 1$ or $\chi_{1,\phi} < 1$ respectively. However, while this $\chi_{1,\phi} = 1$ condition stops the growth of $m_1$ as $L$ grows large, this only tells us about the growth in error vector in expectation. In Pennington et al. (2018), the authors' key motivation is to investigate the possibility of imposing a stronger condition: (approximate) dynamical isometry, where all singular values of the Jacobian are concentrated around one, protecting against even the worst case growth and decay in the error signal.

Equations (28) and (29) yield an expression for the variance of the spectrum of $JJ^\top$ when $\chi_{1,\phi} = 1$,

$$\sigma_{JJ^\top}^2 = m_2 - m_1^2 = L \left( \frac{\mu_1}{\mu_2^2} - 1 - s_1 \right). \tag{30}$$

The variance of the spectrum of $JJ^\top$ for both $\mathsf{ReLU}_\tau$ and $\mathsf{ST}_\tau$ can be readily calculated using (27) in App. F which grows linearly with depth as $L/(1-s)$ and $Ls/(1-s)$ for Gaussian and orthogonal weights respectively.

This shows that even with $\chi_{1,\phi} = 1$, the variance of the Jacobian spectrum grows linearly with depth $L$ for generic $\mu_k$ and $s_1$. Whether or not this is possible to avoid depends on the activation function and weights initialisaton scheme. One case analysed in Pennington et al. (2018), for example, is the HardTanh activation for which $\sigma_w^2(q^*) \longrightarrow 1$ as $q^* \longrightarrow 0$. Together with Equation (30), this means that while for Gaussian weights, with $s_1 = -1$, $\sigma_{JJ^\top}^2 \propto L$ for all $q^*$, with orthogonal weights ($s_1 = 0$) one has that

$$\sigma_{JJ^\top}^2 \longrightarrow 0 \qquad \text{as} \qquad q^* \longrightarrow 0, \tag{31}$$

for fixed $L$, meaning that one can arbitrarily shrink the variance of the spectrum of the Jacobian by sufficiently shrinking $q^*$. They show similarly that it is possible to achieve arbitrarily small $\sigma_{JJ^\top}^2$ for networks with Erf activation functions and orthogonal (but not Gaussian) weights initializations, and also that this is not possible for ReLU networks (neither with Gaussian nor orthogonal weights).

Empirically, even in the absence of dynamical isometry, EoC initialization yields significant benefits in terms of enabling and speeding up initial training of very deep nets Pennington et al. (2017).

The process for calculating EoC $\sigma_w^2$ and $\sigma_b^2$ is as follows: (i) select a $q^*$, say the variance of your (typically already normalised) input data; (ii) use Equation (11) with $\chi_{1,\phi} = 1$ to solve for $\sigma_w^2$; (iii) use Equation (3) with $V(q^*) = q^*$ to solve for $\sigma_b^2$.

## B  VARIANCE AND CORRELATION MAPS FOR $\mathsf{ReLU}_\tau$ AND $\mathsf{ST}_\tau$

Recall that the variance map for a general activation function $\phi$ is the following:

$$V_\phi(q) = \sigma_w^2 \int_{\mathbb{R}} (\phi(\sqrt{q}z))^2 \, \gamma(dz) + \sigma_b^2, \tag{32}$$

where $\gamma(dz) = \exp(-z^2/2)/\sqrt{2\pi}\, dz$ is the standard normal distribution. We define

$$\Phi(z) := \int_{-\infty}^{z} \gamma(dt) \tag{33}$$

being the cumulative distribution function (CDF) of a standard normal distribution, which is related to the error function by

$$\mathrm{erf}(x) = 2\Phi(\sqrt{2}\,x) - 1 \iff \Phi(\sqrt{2}\,x) = \frac{\mathrm{erf}(x) + 1}{2}, \tag{34}$$

$$\mathrm{erf}^{-1}(x) = \frac{1}{\sqrt{2}}\Phi^{-1}\left(\frac{x+1}{2}\right) \iff \Phi^{-1}(x) = \sqrt{2}\mathrm{erf}^{-1}(2x - 1). \tag{35}$$

Recall further that $\Phi(-z) = 1 - \Phi(z)$, as well as the following identities:

$$\int_z^\infty t\,\gamma(dt) = \frac{\exp(-z^2/2)}{\sqrt{2\pi}}, \quad \int_z^\infty t^2\,\gamma(dt) = z\frac{\exp(-z^2/2)}{\sqrt{2\pi}} + (1 - \Phi(z)). \tag{36}$$

With these in mind, the variance map for $\phi = \mathsf{ReLU}_\tau$ can be computed as follows:

$$
\begin{aligned}
V_{\mathsf{ReLU}_\tau}(q) &= \sigma_w^2 \int_{\tau/\sqrt{q}}^\infty (\sqrt{q}z - \tau)^2 \, \gamma(dz) + \sigma_b^2 \\
&= \sigma_w^2 \left[ \int_{\tau/\sqrt{q}}^\infty (qz^2 - 2\sqrt{q}\tau z + \tau^2) \, \gamma(dz) \right] + \sigma_b^2 \\
&= \sigma_w^2 \left[ q\int_{\tau/\sqrt{q}}^\infty z^2 \, \gamma(dz) - 2\sqrt{q}\tau \int_{\tau/\sqrt{q}}^\infty z \, \gamma(dz) + \tau^2 \int_{\tau/\sqrt{q}}^\infty \gamma(dz) \right] + \sigma_b^2 \\
&= \sigma_w^2 \left[ q\left( \frac{\tau}{\sqrt{2\pi q}} \exp\left(-\frac{\tau^2}{2q}\right) + 1 - \Phi\left(\frac{\tau}{\sqrt{q}}\right) \right) \right. \\
&\qquad\left. - \frac{2\sqrt{q}\tau}{\sqrt{2\pi}} \exp\left(-\frac{\tau^2}{2q}\right) + \tau^2 \left( 1 - \Phi\left(\frac{\tau}{\sqrt{q}}\right) \right) \right] + \sigma_b^2 \\
&= \sigma_w^2 \left[ (q + \tau^2)\left( 1 - \Phi\left(\frac{\tau}{\sqrt{q}}\right) \right) - \frac{\sqrt{q}\tau}{\sqrt{2\pi}} \exp\left(-\frac{\tau^2}{2q}\right) \right] + \sigma_b^2. \tag{37}
\end{aligned}
$$

Therefore,

$$
\begin{aligned}
V'_{\mathsf{ReLU}_\tau}(q) &= \sigma_w^2 \left[ \tilde{\Phi}\left(\frac{\tau}{\sqrt{q}}\right) - \frac{\tau(q + \tau^2)}{2q^{3/2}\sqrt{2\pi}} \exp\left(-\frac{\tau^2}{2q}\right) - \frac{\tau}{2\sqrt{2\pi q}} \exp\left(-\frac{\tau^2}{2q}\right) \right. \\
&\qquad\left. - \sqrt{q}\tau\left(\frac{\tau}{\sqrt{2\pi q}}\right) \exp\left(-\frac{\tau^2}{2q}\right) \frac{\tau}{2q^{3/2}} \right] \\
&= \sigma_w^2 \left[ 1 - \Phi\left(\frac{\tau}{\sqrt{q}}\right) \right] = \sigma_w^2 \Phi\left(-\frac{\tau}{\sqrt{q}}\right). \tag{38}
\end{aligned}
$$

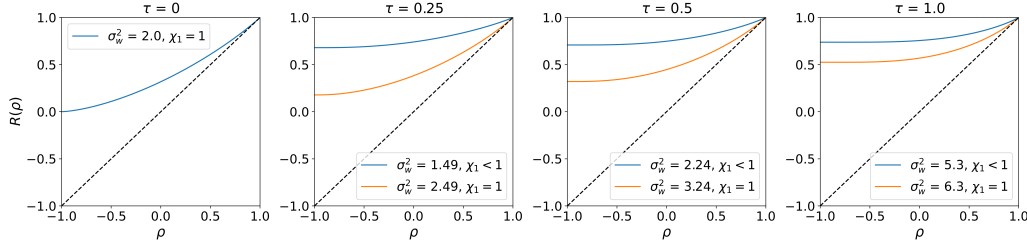

Figure 6: Correlation maps for $\mathsf{ReLU}_\tau$ with $(\sigma_w, \sigma_b)$ on, and on either side of, the EoC, for different values of $\tau$ – except in the case where no $q^*$ exists for those hyperparameters.

Let $q^*$ be a fixed point of the variance map, such that $V_{\mathsf{ReLU}_\tau}(q^*) = q^*$. The slope of the correlation map at $\rho = 1$ is then given by

$$\chi_{1,\mathsf{ReLU}_\tau} = \sigma_w^2 \int_{\mathbb{R}} \left(\phi'(\sqrt{q^*}z)\right)^2 \gamma(dz)$$

$$= \sigma_w^2 \int_{\tau/\sqrt{q^*}}^{\infty} \gamma(dz) = \sigma_w^2 \left[1 - \Phi\left(\frac{\tau}{\sqrt{q^*}}\right)\right] = V'_{\mathsf{ReLU}_\tau}(q^*). \tag{39}$$

As a result, if a DNN is initialized at the EoC (i.e. $\chi_{1,\mathsf{ReLU}_\tau} = 1$), then the fixed point of the variance map satisfies $V'_{\mathsf{ReLU}_\tau}(q^*) = \chi_{1,\mathsf{ReLU}_\tau} = 1$. Moreover, note that for all $q$,

$$V''_{\mathsf{ReLU}_\tau}(q) = \sigma_w^2 \frac{\tau}{2\sqrt{2\pi}q^{3/2}} \exp\left(-\frac{\tau^2}{2q}\right) > 0. \tag{40}$$

This shows that the fixed point $q^*$ is unstable when the DNN is initialized at the EoC.

This phenomenon also holds for DNNs with soft thresholding activation function $\mathsf{ST}_\tau$. In fact,

$$V_{\mathsf{ST}_\tau}(q) = \sigma_w^2 \left[\int_{-\infty}^{-\tau/\sqrt{q}} (\sqrt{q}z + \tau)^2 \gamma(dz) + \int_{\tau/\sqrt{q}}^{\infty} (\sqrt{q}z - \tau)^2 \gamma(dz)\right] + \sigma_b^2$$

$$= 2\sigma_w^2 \left[\int_{\tau/\sqrt{q}}^{\infty} (\sqrt{q}z - \tau)^2 \gamma(dz)\right] + \sigma_b^2$$

$$= 2V_{\mathsf{ReLU}_\tau}(q) - \sigma_b^2 \tag{41}$$

and, if $q^*$ is a fixed point of the above variance map, then the slope of the corresponding correlation map at $\rho = 1$ is

$$\chi_{1,\mathsf{ST}_\tau} = \sigma_w^2 \int_{\mathbb{R}} \left(\phi'(\sqrt{q^*}z)\right)^2 \gamma(dz)$$

$$= \sigma_w^2 \left(\int_{-\infty}^{-\tau/\sqrt{q^*}} + \int_{\tau/\sqrt{q^*}}^{\infty}\right) \gamma(dz) = 2\sigma_w^2 \left[1 - \Phi\left(\frac{\tau}{\sqrt{q^*}}\right)\right]. \tag{42}$$

As a result,

$$V'_{\mathsf{ST}_\tau}(q) = 2V'_{\mathsf{ReLU}_\tau}(q), \quad \chi_{1,\mathsf{ST}_\tau} = 2\chi_{1,\mathsf{ReLU}_\tau} = V'_{\mathsf{ST}_\tau}(q^*), \quad V''_{\mathsf{ST}_\tau}(q) = 2V''_{\mathsf{ReLU}_\tau}(q) > 0, \tag{43}$$

and the fixed point $q^*$ is unstable when such DNN is initialized at the EoC.

Correlation maps $R_\phi(\rho)$ for $\mathsf{ReLU}_\tau$ and $\mathsf{ST}_\tau$ are plotted in Figures 6 and 7 for different choices of the hyperparameters $(\sigma_w, \sigma_b)$, excluding those cases when $q^* > 0$ does not exist. Attempting to regain the stability of $V(q)$ by setting $\chi_{1,\mathsf{ReLU}_\tau} < 1$ would result in the DNN being unstable to small perturbations, i.e. $R(\rho)$ being unstable at $\rho = 1$. Alternatively, having $\chi_{1,\mathsf{ReLU}_\tau} \geq 1$ results in the Gaussian process having variance growing exponentially with depth and the associated exponential scaling of the gradient with depth.

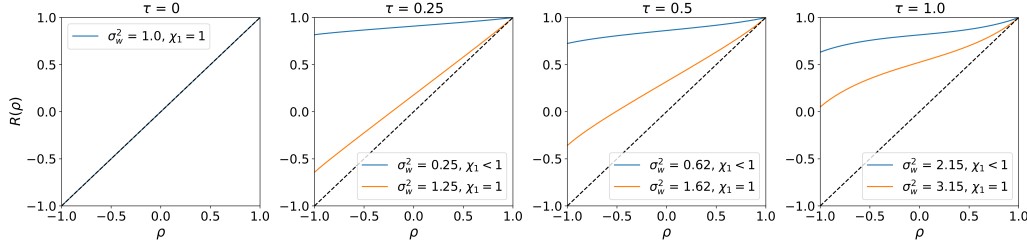

Figure 7: Correlation maps for $\mathsf{ST}_\tau$ with $(\sigma_w, \sigma_b)$ on, and on either side of, the EoC, for different values of $\tau$ – except in the case where no $q^*$ exists for those hyperparameters.

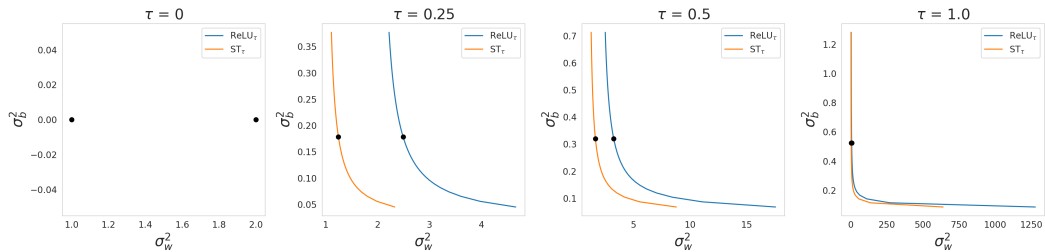

Figure 8: Edge of Chaos for $\mathsf{ReLU}_\tau$ and $\mathsf{ST}_\tau$ for different $\tau$. As expected, the $\sigma_w^2$ values for $\mathsf{ReLU}_\tau$ are twice those for $\mathsf{ST}_\tau$ for the same $q^*$. The black dot corresponds to $q^* = 1$.

## C EXPERIMENTS DEMONSTRATING TRAINING INSTABILITY FOR $\mathsf{ReLU}_\tau$ AND $\mathsf{ST}_\tau$.

Width 300, depth 100 feedforward networks are trained with the above activation functions to perform image classification on the MNIST dataset. The networks are initialized at the EoC using $q^* = 1$, and trained with SGD with $10^{-4}$ learning rate for 200 epochs.

When $\tau > 0$ (the only exception being $\mathsf{ReLU}_\tau$ with $s = 0.5$, i.e. standard ReLU), the input is normalised to have variance $< q^*$, to give the best chance of trainability given the instability of $q^*$ in these cases (in particular the input variance is set to be $0.75$). As seen numerically however, in particular at higher sparsities, these measures are insufficient to enable trainability of $\mathsf{ReLU}_\tau$ and $\mathsf{ST}_\tau$ networks.

A variety of starting sparsity levels ($[50\%, 60\%, 70\%]$) are tested for each activation function, and track both their test accuracy, as well as the sparsity of their activations during evaluation, to check to what extent the initial sparsity levels are maintained during training. We plot the Variance maps $V_\phi$ for each activation function for these settings in Figure 9. For each activation function-hyperparameter combination, we train 5 runs. Table 2 shows the mean and standard deviation of the test accuracy and activation sparsity after training, measured on the test set.

The results concord with our expectations based at the EoC and variance map analyses above. First, $\mathsf{ST}_\tau$ networks fail to train at any of the tested sparsities. This reflects the notable instability of $q^*$ for $\mathsf{ST}_\tau$ with the corresponding values of $\tau$, as shown in Figure 9. Similarly, $\mathsf{ReLU}_\tau$ networks also fail to train once sparsity (and thus $\tau$) grow sufficiently large.

The CNN experiments on the CIFAR10 dataset use networks with 50 layers, and 300 channels per layer, and were trained for 200 epochs with SGD and learning rate 1e-3. A single experiment was run per (activation function, sparsity) pair. Again, both activation functions fail to train at high sparsities.

## D    ADDITIONAL EXPERIMENTAL IMPLEMENTATION DETAILS

For both MNIST and CIFAR10, 10% of the training set was held out as the validation set. Results reported are computed on the test set.

All layers are initialized on the EOC, with a slight modification for the first layer. In our experiments, the first layer is initialized so as to preserve the input variance in the first layer's pre-activations. This was an additional effort to try our best to protect against the instability of $q^*$ for the non-clipped activation functions. The EoC initialization is based on the assumption that the input to the layer are 'post-activations', that is, the incoming vector is a Gaussian which has been passed through the activation function. However, as the input to the first layer has not been passed through an activation function – and thus is dense, not sparse – multiplication of the inputs by a weight matrix initialized at the EoC would substantially grow the variance, possibly to a value larger than $q^*$, which would then prevent convergence to $q^*$ with these activation functions.

Experiments were run on a single V100 GPU, and were implemented using Pytorch Lightning.

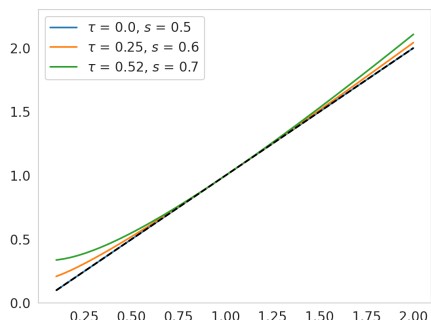 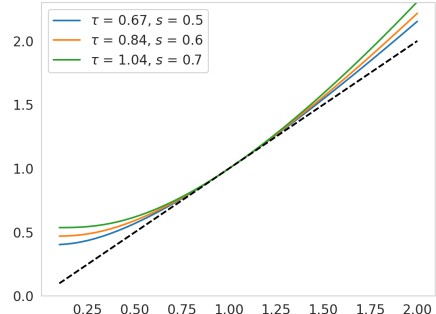

Figure 9: Variance maps for $\mathsf{ReLU}_\tau$ (left) and $\mathsf{ST}_\tau$ (right) with $(\sigma_w, \sigma_b)$ at the EoC for different values of sparsity $s$.

## E    VARIANCE AND CORRELATION MAPS FOR $\mathsf{CReLU}_{\tau,m}$ AND $\mathsf{CST}_{\tau,m}$

The variance map of $\mathsf{CReLU}_{\tau,m}$ is computed as followed:

$$
\begin{aligned}
V_{\mathsf{CReLU}_{\tau,m}}(q) &= \sigma_w^2 \left[ \int_{\tau/\sqrt{q}}^{(\tau+m)/\sqrt{q}} (\sqrt{q}\,z - \tau)^2 \,\gamma(dz) + \int_{(\tau+m)/\sqrt{q}}^{\infty} m^2 \gamma(dz) \right] + \sigma_b^2 \\
&= \sigma_w^2 (v_1(q) - v_2(q) + v_3(q)) + \sigma_b^2,
\end{aligned}
\tag{44}
$$

where

$$
\begin{aligned}
v_1(q) &= \int_{\tau/\sqrt{q}}^{\infty} (\sqrt{q}\,z - \tau)^2 \,\gamma(dz) = \int_{\tau/\sqrt{q}}^{\infty} (qz^2 - 2\sqrt{q}\,\tau z + \tau^2)\,\gamma(dz), \\
v_2(q) &= \int_{(\tau+m)/\sqrt{q}}^{\infty} (\sqrt{q}\,z - \tau)^2 \,\gamma(dz) = \int_{(\tau+m)/\sqrt{q}}^{\infty} (qz^2 - 2\sqrt{q}\,\tau z + \tau^2)\,\gamma(dz), \\
v_3(q) &= \int_{(\tau+m)/\sqrt{q}}^{\infty} m^2 . \gamma(dz).
\end{aligned}
\tag{45}
$$

Notice that $v_1(q) = v_{11}(q) - \tau v_{12}(q) + \tau^2 v_{13}(q)$, where

$$
v_{11}(q) = q \int_{\tau/\sqrt{q}}^{\infty} z^2 \,\gamma(dz), \quad v_{12}(q) = 2\sqrt{q} \int_{\tau/\sqrt{q}}^{\infty} z\,\gamma(dz), \quad v_{13}(q) = \int_{\tau/\sqrt{q}}^{\infty} \gamma(dz).
\tag{46}
$$

Therefore,

$$
\begin{aligned}
\frac{dv_{11}}{dq} &= \frac{d}{dq}\left[q\int_{\tau/\sqrt{q}}^{\infty} z^2\,\gamma(dz)\right] \\
&= \int_{\tau/\sqrt{q}}^{\infty} z^2\,\gamma(dz) + q\frac{d}{dq}\left[\int_{\tau/\sqrt{q}}^{\infty} z^2\,\gamma(dz)\right] \\
&= \frac{\tau}{\sqrt{2\pi q}}\exp\left(-\frac{\tau^2}{2q}\right) + 1 - \Phi\left(\frac{\tau}{\sqrt{q}}\right) + \frac{\tau^3}{2\sqrt{2\pi}\,q^{3/2}}\exp\left(-\frac{\tau^2}{2q}\right), \\
\frac{dv_{12}}{dq} &= \frac{d}{dq}\left[2\sqrt{q}\int_{\tau/\sqrt{q}}^{\infty} z\,\gamma(dz)\right] \\
&= \frac{1}{\sqrt{q}}\int_{\tau/\sqrt{q}}^{\infty} z\,\gamma(dz) + 2\sqrt{q}\frac{d}{dq}\left[\int_{\tau/\sqrt{q}}^{\infty} z\,\gamma(dz)\right] \\
&= \frac{1}{\sqrt{2\pi q}}\exp\left(-\frac{\tau^2}{2q}\right) + \frac{\tau^2}{\sqrt{2\pi}\,q^{3/2}}\exp\left(-\frac{\tau^2}{2q}\right), \\
\frac{dv_{13}}{dq} &= \frac{d}{dq}\left[\int_{\tau/\sqrt{q}}^{\infty}\gamma(dz)\right] = \frac{\tau}{2\sqrt{2\pi}\,q^{3/2}}\exp\left(-\frac{\tau^2}{2q}\right).
\end{aligned}
$$

Hence

$$
\begin{aligned}
\frac{dv_1}{dq} &= \frac{dv_{11}}{dq} - \tau\frac{dv_{12}}{dq} + \tau^2\frac{dv_{13}}{dq} \\
&= \frac{\tau}{\sqrt{2\pi q}}\exp\left(-\frac{\tau^2}{2q}\right) + 1 - \Phi\left(\frac{\tau}{\sqrt{q}}\right) + \frac{\tau^3}{2\sqrt{2\pi}\,q^{3/2}}\exp\left(-\frac{\tau^2}{2q}\right) \\
&\quad - \frac{\tau}{\sqrt{2\pi q}}\exp\left(-\frac{\tau^2}{2q}\right) - \frac{\tau^3}{\sqrt{2\pi}\,q^{3/2}}\exp\left(-\frac{\tau^2}{2q}\right) + \frac{\tau^3}{2\sqrt{2\pi}\,q^{3/2}}\exp\left(-\frac{\tau^2}{2q}\right) \\
&= 1 - \Phi\left(\frac{\tau}{\sqrt{q}}\right).
\end{aligned}
\tag{47}
$$

One could similarly derive

$$
\begin{aligned}
\frac{dv_2}{dq} &= \frac{\tau+m}{\sqrt{2\pi q}}\exp\left(-\frac{(\tau+m)^2}{2q}\right) + 1 - \Phi\left(\frac{\tau+m}{\sqrt{q}}\right) + \frac{(\tau+m)^3}{2\sqrt{2\pi}\,q^{3/2}}\exp\left(-\frac{(\tau+m)^2}{2q}\right) \\
&\quad - \frac{\tau}{\sqrt{2\pi q}}\exp\left(-\frac{(\tau+m)^2}{2q}\right) - \frac{\tau(\tau+m)^2}{\sqrt{2\pi}\,q^{3/2}}\exp\left(-\frac{(\tau+m)^2}{2q}\right) \\
&\quad + \frac{\tau^2(\tau+m)}{2\sqrt{2\pi}\,q^{3/2}}\exp\left(-\frac{(\tau+m)^2}{2q}\right) \\
&= \frac{m}{\sqrt{2\pi q}}\exp\left(-\frac{(\tau+m)^2}{2q}\right) + 1 - \Phi\left(\frac{\tau+m}{\sqrt{q}}\right) + \frac{m^2(\tau+m)}{2\sqrt{2\pi}\,q^{3/2}}\exp\left(-\frac{(\tau+m)^2}{2q}\right)
\end{aligned}
\tag{48}
$$

Finally,

$$
\frac{dv_3}{dq} = \frac{m^2(\tau+m)}{2\sqrt{2\pi}\,q^{3/2}}\exp\left(-\frac{(\tau+m)^2}{2q}\right),
\tag{49}
$$

so

$$
\begin{aligned}
\frac{dV_{\mathsf{CReLU}_{\tau,m}}}{dq} &= \sigma_w^2\left[\frac{dv_1}{dq} - \frac{dv_2}{dq} + \frac{dv_3}{dq}\right] \\
&= \sigma_w^2\left[\Phi\left(\frac{\tau+m}{\sqrt{q}}\right) - \Phi\left(\frac{\tau}{\sqrt{q}}\right) - \frac{m}{\sqrt{2\pi q}}\exp\left(-\frac{(\tau+m)^2}{2q}\right)\right] \\
&= \sigma_w^2\left[\frac{1}{2}\mathrm{erf}\left(\frac{\tau+m}{\sqrt{2q}}\right) - \frac{1}{2}\mathrm{erf}\left(\frac{\tau}{\sqrt{2q}}\right) - \frac{m}{\sqrt{2\pi q}}\exp\left(-\frac{(\tau+m)^2}{2q}\right)\right].
\end{aligned}
\tag{50}
$$

The slope of the associated correlation map at $\rho = 1$ for DNNs initialized at the fixed point $q^*$ is

$$\chi_{1,\text{CReLU}_{\tau,m}} = \sigma_w^2 \int_{\tau/\sqrt{q^*}}^{(\tau+m)/\sqrt{q^*}} \gamma(dz) = \sigma_w^2 \left( \Phi\left(\frac{\tau+m}{\sqrt{q^*}}\right) - \Phi\left(\frac{\tau}{\sqrt{q^*}}\right) \right)$$

$$= \sigma_w^2 \left[ \frac{1}{2}\text{erf}\left(\frac{\tau+m}{\sqrt{2q^*}}\right) - \frac{1}{2}\text{erf}\left(\frac{\tau}{\sqrt{2q^*}}\right) \right]. \qquad (51)$$

Therefore

$$\chi_{1,\text{CReLU}_{\tau,m}} = V'_{\text{CReLU}_{\tau,m}}(q^*) + \frac{\sigma_w^2 m}{\sqrt{2\pi q^*}} \exp\left(-\frac{(m+\tau)^2}{2q^*}\right) > V'_{\text{CReLU}_{\tau,m}}(q^*), \qquad (52)$$

which makes any fixed points $q^*$ of $V_{\text{CReLU}_{\tau,m}}$ at the EoC locally stable, as described in Sec. 3.

For the other activation functions $\text{CST}_{\tau,m}$, the variance map is:

$$V_{\text{CST}_{\tau,m}}(q) = \sigma_w^2 \left[ \int_{-\infty}^{-(\tau+m)/\sqrt{q}} m^2 \, \gamma(dz) + \int_{-(\tau+m)/\sqrt{q}}^{-\tau/\sqrt{q}} (\sqrt{q}z + \tau)^2 \, \gamma(dz) \right.$$

$$\left. + \int_{\tau/\sqrt{q}}^{(\tau+m)/\sqrt{q}} (\sqrt{q}z - \tau)^2 \, \gamma(dz) + \int_{(\tau+m)/\sqrt{q}}^{\infty} m^2 \, \gamma(dz) \right] + \sigma_b^2$$

$$= 2\sigma_w^2 \left[ \int_{\tau/\sqrt{q}}^{(\tau+m)/\sqrt{q}} (\sqrt{q}z - \tau)^2 \, \gamma(dz) + \int_{(\tau+m)/\sqrt{q}}^{\infty} m^2 \, \gamma(dz) \right] + \sigma_b^2$$

$$= 2V_{\text{CReLU}_{\tau,m}}(q) - \sigma_b^2, \qquad (53)$$

and the slope of the associated correlation map is

$$\chi_{1,\text{CST}_{\tau,m}} = \sigma_w^2 \left( \int_{-(\tau+m)/\sqrt{q^*}}^{-\tau/\sqrt{q^*}} + \int_{\tau/\sqrt{q^*}}^{(\tau+m)/\sqrt{q^*}} \right) \gamma(dz)$$

$$= 2\sigma_w^2 \int_{\tau/\sqrt{q^*}}^{(\tau+m)/\sqrt{q^*}} \gamma(dz) = 2\chi_{1,\text{CReLU}_{\tau,m}}. \qquad (54)$$

This yields $V'_{\text{CST}_{\tau,m}}(q^*) = 2V'_{\text{CReLU}_{\tau,m}}(q^*)$, and that

$$\chi_{1,\text{CST}_{\tau,m}} = 2\chi_{1,\text{CReLU}_{\tau,m}} = V'_{\text{CST}_{\tau,m}}(q^*) + \frac{\sqrt{2}\,\sigma_w^2 m}{\sqrt{\pi q^*}} \exp\left(-\frac{(m+\tau)^2}{2q^*}\right) > V'_{\text{CST}_{\tau,m}}(q^*), \quad (55)$$

which shows that any fixed points $q^*$ of $V_{\text{CST}_{\tau,m}}$ at the EoC are locally stable as well.

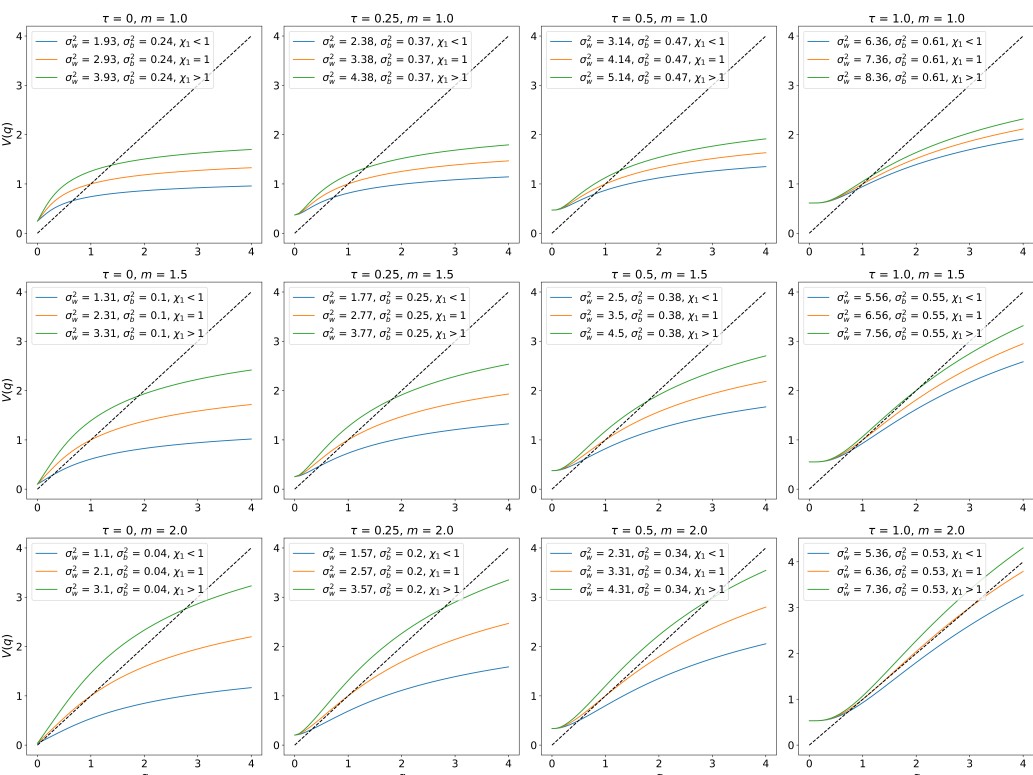

Figure 10: Variance maps for CReLU$_{\tau,m}$ with $(\sigma_w, \sigma_b)$ at and on either side of the EoC, for different $s$ and $m$, with $q^* = 1$. From top to bottom, the rows correspond to $m = 1$, $m = 1.5$, and $m = 2$, and from left to right the columns correspond to $\tau = 0$, $\tau = 0.25$, $\tau = 0.5$ and $\tau = 1$.

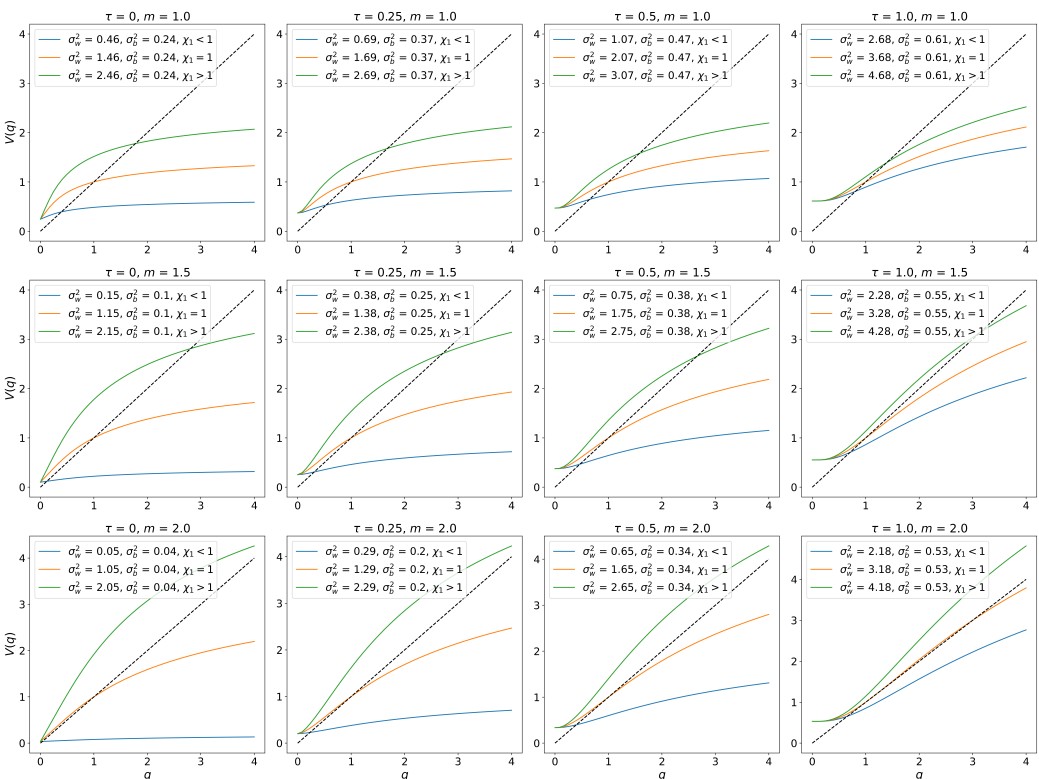

Figure 11: Variance maps for $\mathsf{CST}_{\tau,m}$ with $(\sigma_w, \sigma_b)$ at and on either side of the EoC, for different $\tau$ and $m$, with $q^* = 1$. From top to bottom, the rows correspond to $m = 1$, $m = 1.5$, and $m = 2$, and from left to right the columns correspond to $\tau = 0$, $\tau = 0.25$, $\tau = 0.5$ and $\tau = 1$.

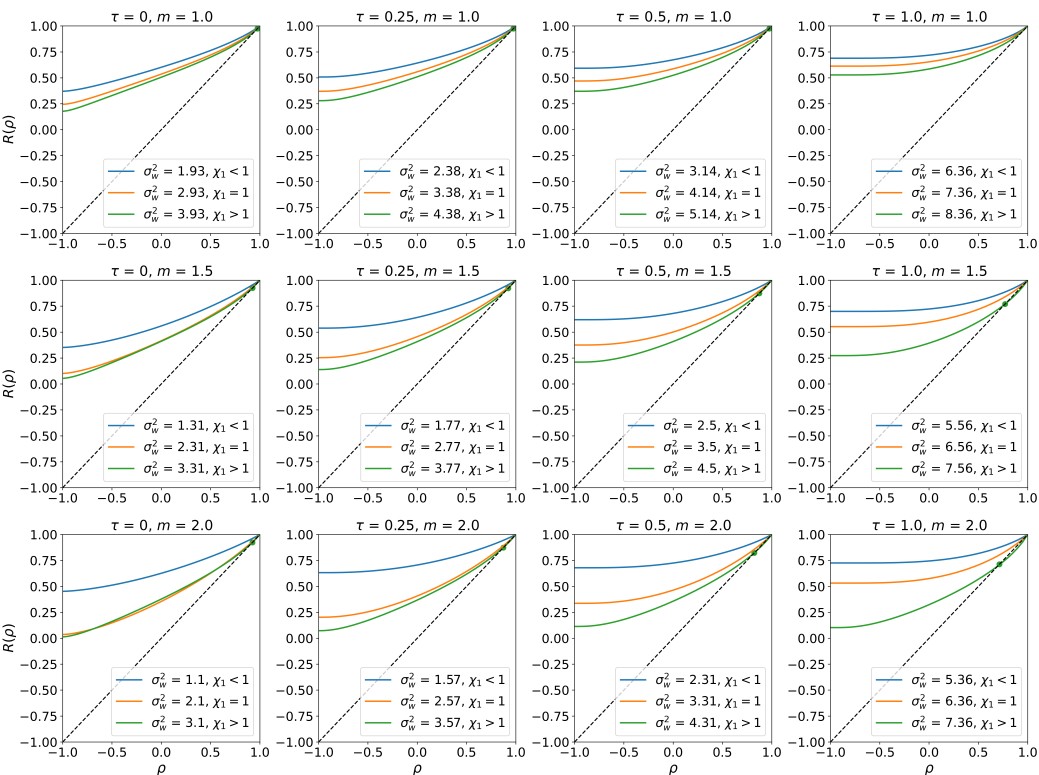

Figure 12: Correlation maps for $\mathsf{CReLU}_{\tau,m}$ with $(\sigma_w, \sigma_b)$ at and on either side of the EoC, for different $\tau$ and $m$. From top to bottom, the rows correspond to $m = 1$, $m = 1.5$, and $m = 2$, and from left to right the columns correspond to $\tau = 0$, $\tau = 0.25$, $\tau = 0.5$ and $\tau = 1$. Fixed points $\rho^* < 1$ (where the correlation map crosses the diagonal $R(\rho) = \rho$) are indicated by round markers on the relevant curves.

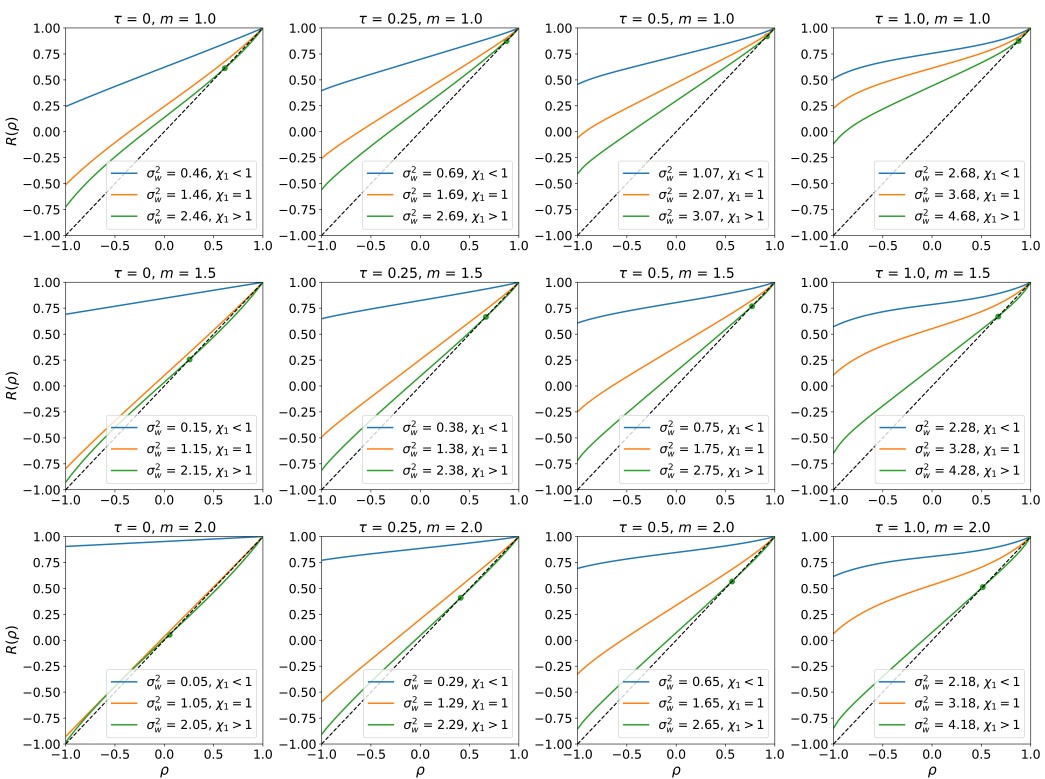

Figure 13: Correlation maps for $\mathrm{CST}_{\tau,m}$ with $(\sigma_w, \sigma_b)$ at and on either side of the EoC, for different $\tau$ and $m$. From top to bottom, the rows correspond to $m = 1$, $m = 1.5$, and $m = 2$, and from left to right the columns correspond to $\tau = 0$, $\tau = 0.25$, $\tau = 0.5$ and $\tau = 1$. Fixed points $\rho^* < 1$ (where the correlation map crosses the diagonal $R(\rho) = \rho$) are indicated by round markers on the relevant curves.

## F  RELATIONSHIP BETWEEN $\sigma_w^2$ AND THE EXPECTED SPARSITY AT THE EOC, CONTROL OF VARIANCE OF THE INPUT-OUTPUT JACOBIAN.

Recall that the 'expected sparsity' with respect to an activation function $\phi$ could be computed under the usual mean-field assumption (that each pre-activation $Z$ follows a normal distribution with variance $q^*$):

$$s = \mathbb{P}(\phi(Z) = 0), \quad Z \sim \mathcal{N}(0, q^*). \tag{56}$$

When studying DNNs with sparsifying activation functions ($\phi = \mathsf{ReLU}_\tau$, $\mathsf{ST}_\tau$, $\mathsf{CReLU}_{\tau,m}$ or $\mathsf{CST}_{\tau,m}$), we would like to find the optimal $\hat{\tau}$ such that the DNNs achieve the expected sparsity $s$. The optimal $\tau$ is related to the $s$ and $q^*$ as followed:

- For $\phi = \mathsf{ReLU}_\tau$ or $\mathsf{CReLU}_{\tau,m}$:

$$\begin{aligned}
\hat{\tau} = \hat{\tau}_\phi(s, q^*) &:= \inf_\tau \{\tau \mid \mathbb{P}(\phi(Z) = 0) \geq s, \, Z \sim \mathcal{N}(0, q^*)\} \\
&= \inf_\tau \{\tau \mid \mathbb{P}(Z < \tau) \geq s, \, Z \sim \mathcal{N}(0, q^*)\} \\
&= \sqrt{q^*}\Phi^{-1}(s).
\end{aligned} \tag{57}$$

- and for $\phi = \mathsf{ST}_\tau$ or $\mathsf{CST}_{\tau,m}$,

$$\begin{aligned}
\hat{\tau} = \hat{\tau}_\phi(s, q^*) &:= \inf_\tau \{\tau \mid \mathbb{P}(\phi(Z) = 0) \geq s, \, Z \sim \mathcal{N}(0, q^*)\} \\
&= \inf_\tau \{\tau \mid \mathbb{P}(-\tau < Z < \tau) \geq s, \, Z \sim \mathcal{N}(0, q^*)\} \\
&= \sqrt{2q^*}\mathrm{erf}^{-1}(s) = \sqrt{q^*}\Phi^{-1}\left(\frac{s+1}{2}\right).
\end{aligned} \tag{58}$$

There is a simple formula to compute $\sigma_w^2$ for initialising the DNNs at the EoC ($\chi_{1,\phi} = 1$) with activation functions $\phi = \mathsf{ReLU}_\tau$ or $\mathsf{ST}_\tau$. In particular,

- for $\phi = \mathsf{ReLU}_\tau$:

$$\sigma_w^2 = \sigma_w^2(s, q^*) := \left(1 - \Phi\left(\frac{\hat{\tau}_\phi(s, q^*)}{\sqrt{q^*}}\right)\right)^{-1} = \left(1 - \Phi\left(\Phi^{-1}(s)\right)\right)^{-1} = \frac{1}{1-s}, \tag{59}$$

- and for $\mathsf{ST}_\tau$, using the identity $\mathrm{erf}^{-1}(s) = \frac{1}{\sqrt{2}}\Phi^{-1}\left(\frac{s+1}{2}\right)$:

$$\begin{aligned}
\sigma_w^2 = \sigma_w^2(s, q^*) &:= \frac{1}{2}\left(1 - \Phi\left(\frac{\hat{\tau}(s, q^*)}{\sqrt{q^*}}\right)\right)^{-1} \\
&= \frac{1}{2}\left(1 - \Phi\left(\Phi^{-1}\left(\frac{s+1}{2}\right)\right)\right)^{-1} = \frac{1}{1-s}.
\end{aligned} \tag{60}$$

Notice that the $\sigma_w^2$ above are independent of the variance of the pre-activation $q^*$. This is not true for the cases of the clipped activations $\phi = \mathsf{CReLU}_{\tau,m}$ or $\mathsf{CST}_{\tau,m}$, instead, the initialization $\sigma_w^2$ at the EoC now depends on $q^*$, $m$, and $s$. In fact,

- for $\mathsf{CReLU}_{\tau,m}$, using the identity $\Phi^{-1}(s) = \sqrt{2}\mathrm{erf}^{-1}(2s-1)$:

$$\begin{aligned}
\sigma_w^2 &= 2\left(\mathrm{erf}\left(\frac{m+\tau}{\sqrt{2q^*}}\right) - \mathrm{erf}\left(\frac{\tau}{\sqrt{2q^*}}\right)\right)^{-1} \\
&= 2\left(\mathrm{erf}\left(\frac{m+\sqrt{2q^*}\mathrm{erf}^{-1}(2s-1)}{\sqrt{2q^*}}\right) - \mathrm{erf}\left(\mathrm{erf}^{-1}(2s-1)\right)\right)^{-1} \\
&= 2\left(\mathrm{erf}\left(\frac{m}{\sqrt{2q^*}} + \mathrm{erf}^{-1}(2s-1)\right) - 2s + 1\right)^{-1},
\end{aligned} \tag{61}$$

- and for $\text{CST}_{\tau,m}$:

$$
\begin{aligned}
\sigma_w^2 &= \left( \text{erf}\left( \frac{m+\tau}{\sqrt{2q^*}} \right) - \text{erf}\left( \frac{\tau}{\sqrt{2q^*}} \right) \right)^{-1} \\
&= \left( \text{erf}\left( \frac{m + \sqrt{2q^*}\text{erf}^{-1}(s)}{\sqrt{2q^*}} \right) - \text{erf}\left( \text{erf}^{-1}(s) \right) \right)^{-1} \\
&= \left( \text{erf}\left( \frac{m}{\sqrt{2q^*}} + \text{erf}^{-1}(s) \right) - s \right)^{-1}.
\end{aligned}
\tag{62}
$$

Despite the fact that that $\sigma_w^2$ is now dependent on $q^*$, the derivatives of $\text{CReLU}_{\tau,m}$ and $\text{CST}_{\tau,m}$ take only values of zero or one, like those of their non-clipped counterparts, and thus the moments of the spectra as computed by (27) are independent of $k$, see Table 3. Once again, for both clipped activation functions, $\mu_k = \mu$ for all $k$ such that $\mu_1/\mu_2^2 = \sigma_w^2$ as was the case for $\text{ReLU}_\tau$ and $\text{ST}_\tau$. For both $\text{CReLU}_{\tau,m}$ ($s \geq 0.5$) and $\text{CST}_{\tau,m}$ ($s \geq 0$), from Equations (61) and (62) that

$$
\sigma_w^2 \longrightarrow \frac{1}{1-s} \qquad \text{as} \qquad q^* \longrightarrow 0,
\tag{63}
$$

meaning that $\sigma_{JJ^\top}^2$ grows linearly with $L$ with these activation functions, with at least the same rate as their unclipped counterparts.

Nonetheless, the prospect of a stable EoC initialization stands as a substantial improvement over the unclipped variants, and as we show in Section 4, this appears sufficient to enable trainability of very deep networks with sparse activations in practice.

Table 3: Summary statistics for the Jacobian's spectrum

| | $\mu_k\ (=\mu)$ | $\sigma_w^2$ | $M_{D^2}(z)$ | $\sigma_{JJ^\top}^2$ |
|---|---|---|---|---|
| $\text{ReLU}_\tau$ | $1 - \Phi\left( \frac{\tau}{\sqrt{q^*}} \right)$ | $\frac{1}{\mu}$ | $\frac{1}{\sigma_w^2}\frac{1}{1-z}$ | $L\left( \sigma_w^2 - 1 - s_1 \right)$ |
| $\text{ST}_\tau$ | $2\left( 1 - \Phi\left( -\frac{\tau}{\sqrt{q^*}} \right) \right)$ | $\frac{1}{\mu}$ | $\frac{1}{\sigma_w^2}\frac{1}{1-z}$ | $L\left( \sigma_w^2 - 1 - s_1 \right)$ |
| $\text{CReLU}_{\tau,m}$ | $\frac{1}{2}\left( \text{erf}\left( \frac{m+\tau}{\sqrt{2q^*}} \right) - \text{erf}\left( \frac{\tau}{\sqrt{2q^*}} \right) \right)$ | $\frac{1}{\mu}$ | $\frac{1}{\sigma_w^2}\frac{1}{1-z}$ | $L\left( \sigma_w^2 - 1 - s_1 \right)$ |
| $\text{CST}_{\tau,m}$ | $\text{erf}\left( \frac{m+\tau}{\sqrt{2q^*}} \right) - \text{erf}\left( \frac{\tau}{\sqrt{2q^*}} \right)$ | $\frac{1}{\mu}$ | $\frac{1}{\sigma_w^2}\frac{1}{1-z}$ | $L\left( \sigma_w^2 - 1 - s_1 \right)$ |

## G FAILURE TO TRAIN FOR HIGH SPARSITY AND LOW MAGNITUDE CLIPPING

The second failure mode discussed in Section 4 appears not to be a failure of the initialization, and thus rather suggests the existence of challenges when training networks with such a high level of activation sparsity combined with small values of $m$. Indeed, inspecting the training loss curves (see e.g. Figure 14) suggests that though training begins in an expected fashion, the problem for those runs exhibiting low accuracy is a combination of slow training, and instability later on in training—in these cases, we see the emergence of large spikes in training loss, which (to varying degrees) are not fully recovered from. Investigating the source and/or dynamics of the observed training speed and stability issues late in training is beyond the scope of this paper, and we defer it to future work.

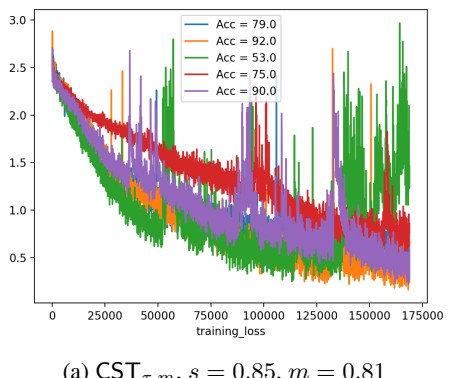
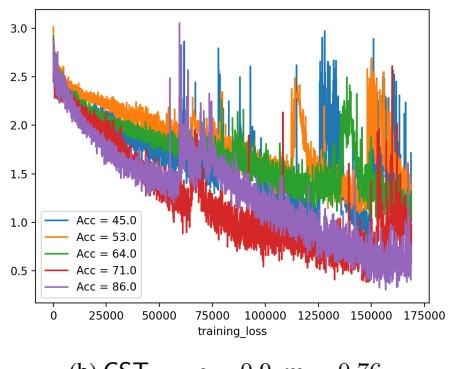

(a) $CST_{\tau,m}$, $s = 0.85$, $m = 0.81$     (b) $CST_{\tau,m}$, $s = 0.9$, $m = 0.76$

Figure 14: The loss curves during training for each of the 5 runs, for two activation function and hyperparameter configurations with high activation sparsity which achieve low test accuracy as a result of slow and/or unstable training. The legend shows the corresponding final test accuracy achieved by each model.

## H EXPERIMENTS WITH SHALLOWER NETWORKS

We have repeat the DNN experiments on MNIST from the main paper, but with depth 30 instead of depth 100. The results are shown in Tables 4 and 5

We can see from these new results that the key conclusions from the 100 layer experiments hold true with 30 layers too. While $ReLU_\tau$ and $ST_\tau$ networks fail to train consistently once sparsity reaches 80% and 70% respectively, $CReLU_{\tau,m}$ and $CST_{\tau,m}$ networks can train consistently to high accuracy with activation sparsity of 90%.

|  | $s$ | $\tau$ | Test Accuracy mean | std | Test Sparsity mean | std |
|---|---|---|---|---|---|---|
| ReLU$_\tau$ | 0.70 | 0.52 | 0.93 | 0.01 | 0.67 | 0.01 |
|  | 0.80 | 0.84 | 0.60 | 0.46 | 0.47 | 0.42 |
|  | 0.85 | 1.04 | 0.27 | 0.37 | 0.18 | 0.38 |
|  | 0.90 | 1.28 | 0.10 | 0.00 | 0.02 | 0.00 |
| ST$_\tau$ | 0.70 | 1.04 | 0.27 | 0.38 | 0.14 | 0.31 |
|  | 0.80 | 1.28 | 0.10 | 0.00 | 0.00 | 0.00 |
|  | 0.85 | 1.44 | 0.10 | 0.00 | 0.01 | 0.00 |
|  | 0.90 | 1.64 | 0.27 | 0.37 | 0.19 | 0.38 |

Table 4: Experimental results for ReLU$_\tau$ and ST$_\tau$ DNNs with 30 layers on MNIST, with sparsity $s$ up to 0.9. The mean and standard deviation of 5 runs are reported for both test accuracy and average activation sparsity calculated on the test set.

|  | $s$ | $\tau$ | $m$ | $V'(q^*)$ | $V''(q^*)$ | Test Accuracy mean | std | Test Sparsity mean | std |
|---|---|---|---|---|---|---|---|---|---|
| CReLU$_{\tau,m}$ | 0.70 | 0.52 | 1.05 | 0.5 | -0.37 | 0.91 | 0.01 | 0.70 | 0.01 |
|  |  |  | 1.45 | 0.7 | -0.31 | 0.92 | 0.00 | 0.70 | 0.01 |
|  |  |  | 2.05 | 0.9 | -0.04 | 0.93 | 0.01 | 0.69 | 0.01 |
|  | 0.80 | 0.84 | 0.89 | 0.5 | -0.24 | 0.90 | 0.02 | 0.80 | 0.01 |
|  |  |  | 1.27 | 0.7 | -0.12 | 0.92 | 0.01 | 0.80 | 0.01 |
|  |  |  | 1.85 | 0.9 | 0.21 | 0.92 | 0.01 | 0.78 | 0.02 |
|  | 0.85 | 1.04 | 0.81 | 0.5 | -0.14 | 0.90 | 0.01 | 0.85 | 0.01 |
|  |  |  | 1.17 | 0.7 | 0.02 | 0.91 | 0.01 | 0.84 | 0.01 |
|  |  |  | 1.74 | 0.9 | 0.41 | 0.92 | 0.01 | 0.84 | 0.01 |
|  | 0.90 | 1.28 | 0.72 | 0.5 | 0.00 | 0.85 | 0.06 | 0.90 | 0.01 |
|  |  |  | 1.06 | 0.7 | 0.23 | 0.90 | 0.03 | 0.89 | 0.01 |
|  |  |  | 1.61 | 0.9 | 0.69 | 0.91 | 0.01 | 0.74 | 0.02 |
| CST$_{\tau,m}$ | 0.70 | 1.04 | 0.81 | 0.5 | -0.14 | 0.91 | 0.01 | 0.70 | 0.00 |
|  |  |  | 1.17 | 0.7 | 0.02 | 0.92 | 0.01 | 0.69 | 0.01 |
|  |  |  | 1.74 | 0.9 | 0.41 | 0.93 | 0.01 | 0.66 | 0.02 |
|  | 0.80 | 1.28 | 0.72 | 0.5 | 0.00 | 0.90 | 0.01 | 0.80 | 0.01 |
|  |  |  | 1.06 | 0.7 | 0.23 | 0.91 | 0.01 | 0.79 | 0.01 |
|  |  |  | 1.61 | 0.9 | 0.69 | 0.92 | 0.01 | 0.49 | 0.16 |
|  | 0.85 | 1.44 | 0.67 | 0.5 | 0.11 | 0.88 | 0.02 | 0.84 | 0.01 |
|  |  |  | 1.00 | 0.7 | 0.39 | 0.91 | 0.01 | 0.84 | 0.01 |
|  |  |  | 1.53 | 0.9 | 0.89 | 0.27 | 0.37 | 0.44 | 0.21 |
|  | 0.90 | 1.64 | 0.62 | 0.5 | 0.28 | 0.77 | 0.17 | 0.90 | 0.01 |
|  |  |  | 0.93 | 0.7 | 0.63 | 0.91 | 0.01 | 0.89 | 0.01 |
|  |  |  | 1.44 | 0.9 | 1.20 | 0.11 | 0.00 | 0.25 | 0.01 |

Table 5: Experimental results for CReLU$_{\tau,m}$ and CST$_{\tau,m}$ DNNs wth 30 layers on MNIST, with sparsity $s$ up to 0.9. The mean and standard deviation of 5 runs are reported for both test accuracy and average activation sparsity calculated on the test set.

