# OpenReview forum: "DEEP NEURAL NETWORK INITIALIZATION WITH SPARSITY INDUCING ACTIVATIONS"
_ICLR.cc/2024/Conference — ICLR 2024 poster_

### Official Review · Reviewer_FeGe · 2023-10-30

**Soundness:** 4 excellent
**Presentation:** 3 good
**Contribution:** 2 fair
**Rating:** 6
**Confidence:** 4

**Summary:**

The paper studies how very deep neural networks, including densely connected and convolutional networks, behave at initialization when using sparsity inducing activation functions. The two natural sparsity-inducing functions studied in the paper are the shifted ReLU activation, which is just a ReLU with a fixed bias, and the soft thresholding function, an activation that evaluates to zero in some fixed interval. The main result shows that these activations make the initialization unstable for very deep networks. This instability can be fixed by using a clipped version of these activation functions. The authors show some experiments, demonstrating that deep networks can be trained with a clipped version of the above activation functions, with minor drop in accuracy.

**Strengths:**

Using sparse networks, with sparse activations and/or sparse weights, is an important field of study, and it seems that the paper gives some contributions on how the activation function of the network should be adapted to support training networks with sparse activations. This work can potentially serve as a basis for future works on building sparse neural networks.
The theoretical analysis of sparse activation functions, their problematic behavior at initialization of deep networks and the solution of clipping the weights is to my understanding novel.

**Weaknesses:**

The main weakness I find in the paper is that while the motivation for the paper comes from a practical perspective, namely building neural networks with sparse activations that can be implemented more efficiently in practice, it seems that the applicability of the results is not clear. To my understanding, the results only apply for very deep neural networks (the experiments use 100-layer networks). The authors should clarify whether or not their results apply to networks of reasonable depth. Specifically, it would be good to show some experiment for networks of reasonable depth and show how the activation choice affects the behavior. It seems that in this setting depth is only hurting performance, so while it is in theory interesting to analyze how such deep networks should be trained, it seems that the applicability of this method is limited.

The authors should also discuss the effect of adding residual connections on the stability of the network training. As residual-networks has been the main solution for training very deep networks, the authors should clarify whether their results also apply for residual networks.

Additionally, it seems that some of the experiments were left out of the main paper, and only appear in the appendix (for example, studying the CST activation and comparing the clipped activations with non-clipped ones). These are key experiments in the paper and should appear in the main text.

**Questions:**

See above.

==================

Score is updated after author response.

---

> ### Author Response · Authors · 2023-11-15
> **Authors' response**
>
> We are very grateful to the reviewer for their careful reading of our paper, thoughtful feedback, and constuctive suggestions. Below we respond to the concerns and/or questions raised.
>
> ___
>
> **Comment:**
>
> The main weakness I find in the paper is that while the motivation for the paper comes from a practical perspective, namely building neural networks with sparse activations that can be implemented more efficiently in practice, it seems that the applicability of the results is not clear. To my understanding, the results only apply for very deep neural networks (the experiments use 100-layer networks). The authors should clarify whether or not their results apply to networks of reasonable depth. Specifically, it would be good to show some experiment for networks of reasonable depth and show how the activation choice affects the behavior. It seems that in this setting depth is only hurting performance, so while it is in theory interesting to analyze how such deep networks should be trained, it seems that the applicability of this method is limited.
>
> **Response:**
>
> While we agree that our paper provides primarily a theoretical contribution -- insofar as the experiments presented are on relatively simple tasks, and were primarily designed to verify the theory -- we believe that the contributions are important and of interest to the community nonetheless. The key goal of EoC initialisations is to preserve signal propagation and avoid vanishing and exploding gradients, and the best proof of the ability to do this is to show that we can train networks with very many layers. This is what motivated our choice of experiments. We should also note that our work is not unique in this regard, and similar experimental setups are relatively common in experiments which develop EoC theory.
>
> Having said that, we think that your comment is fair and important, and that it is worthwhile to test how the comparisons play out in practice when the networks are not quite so deep. To explore this, we have repeated the DNN experiments from the paper, but with depth=30 instead of depth=100. The results are shown in the table below, and are now included in Table 4 and 5, in Appendix H.

---

> > ### Author Response · Authors · 2023-11-15
> > **Authors' response part 2**
> >
> > **Experimental results with 30 layers DNNs:**
> >
> > |             |      |        |      | |  |   |
> > |-------------|------|--------|------|------|------|------|
> > |             |      |        | mean accuracy | accuracy std  | mean sparsity| sparsity std  |
> > |             | $s$  | $\tau$ |      |      |      |      |
> > | ReLU$_\tau$ | 0.70 | 0.52   | 0.93 | 0.01 | 0.67 | 0.01 |
> > |             | 0.80 | 0.84   | 0.60 | 0.46 | 0.47 | 0.42 |
> > |             | 0.85 | 1.04   | 0.27 | 0.37 | 0.18 | 0.38 |
> > |             | 0.90 | 1.28   | 0.10 | 0.00 | 0.02 | 0.00 |
> > | ST$_\tau$   | 0.70 | 1.04   | 0.27 | 0.38 | 0.14 | 0.31 |
> > |             | 0.80 | 1.28   | 0.10 | 0.00 | 0.00 | 0.00 |
> > |             | 0.85 | 1.44   | 0.10 | 0.00 | 0.01 | 0.00 |
> > |             | 0.90 | 1.64   | 0.27 | 0.37 | 0.19 | 0.38 |
> >
> >
> >
> >
> >
> > |          |      |        |      |           |            | mean accuracy| accuracy std  | mean sparsity | sparsity std  |
> > |----------|------|--------|------|-----------|------------|------|------|------|------|
> > |          | $s$  | $\tau$ | $m$  | $V'(q^*)$ | $V''(q^*)$ |      |      |      |      |
> > | CReLU$_\tau$ | 0.70 | 0.52   | 1.05 | 0.5       | -0.37      | 0.91 | 0.01 | 0.70 | 0.01 |
> > |          |      |        | 1.45 | 0.7       | -0.31      | 0.92 | 0.00 | 0.70 | 0.01 |
> > |          |      |        | 2.05 | 0.9       | -0.04      | 0.93 | 0.01 | 0.69 | 0.01 |
> > |          | 0.80 | 0.84   | 0.89 | 0.5       | -0.24      | 0.90 | 0.02 | 0.80 | 0.01 |
> > |          |      |        | 1.27 | 0.7       | -0.12      | 0.92 | 0.01 | 0.80 | 0.01 |
> > |          |      |        | 1.85 | 0.9       | 0.21       | 0.92 | 0.01 | 0.78 | 0.02 |
> > |          | 0.85 | 1.04   | 0.81 | 0.5       | -0.14      | 0.90 | 0.01 | 0.85 | 0.01 |
> > |          |      |        | 1.17 | 0.7       | 0.02       | 0.91 | 0.01 | 0.84 | 0.01 |
> > |          |      |        | 1.74 | 0.9       | 0.41       | 0.92 | 0.01 | 0.84 | 0.01 |
> > |          | 0.90 | 1.28   | 0.72 | 0.5       | 0.00       | 0.85 | 0.06 | 0.90 | 0.01 |
> > |          |      |        | 1.06 | 0.7       | 0.23       | 0.90 | 0.03 | 0.89 | 0.01 |
> > |          |      |        | 1.61 | 0.9       | 0.69       | 0.91 | 0.01 | 0.74 | 0.02 |
> > | CST$_\tau$   | 0.70 | 1.04   | 0.81 | 0.5       | -0.14      | 0.91 | 0.01 | 0.70 | 0.00 |
> > |          |      |        | 1.17 | 0.7       | 0.02       | 0.92 | 0.01 | 0.69 | 0.01 |
> > |          |      |        | 1.74 | 0.9       | 0.41       | 0.93 | 0.01 | 0.66 | 0.02 |
> > |          | 0.80 | 1.28   | 0.72 | 0.5       | 0.00       | 0.90 | 0.01 | 0.80 | 0.01 |
> > |          |      |        | 1.06 | 0.7       | 0.23       | 0.91 | 0.01 | 0.79 | 0.01 |
> > |          |      |        | 1.61 | 0.9       | 0.69       | 0.92 | 0.01 | 0.49 | 0.16 |
> > |          | 0.85 | 1.44   | 0.67 | 0.5       | 0.11       | 0.88 | 0.02 | 0.84 | 0.01 |
> > |          |      |        | 1.00 | 0.7       | 0.39       | 0.91 | 0.01 | 0.84 | 0.01 |
> > |          |      |        | 1.53 | 0.9       | 0.89       | 0.27 | 0.37 | 0.44 | 0.21 |
> > |          | 0.90 | 1.64   | 0.62 | 0.5       | 0.28       | 0.77 | 0.17 | 0.90 | 0.01 |
> > |          |      |        | 0.93 | 0.7       | 0.63       | 0.91 | 0.01 | 0.89 | 0.01 |
> > |          |      |        | 1.44 | 0.9       | 1.20       | 0.11 | 0.00 | 0.25 | 0.01 |
> >
> >
> > We can see from these new results that the key conclusions from the 100 layer experiments hold true with 30 layers too. While $ReLU_\tau$ and $ST_\tau$ networks fail to train consistently once sparsity reaches 80\% and 70\% respectively, $CReLU_{\tau, m}$ and $CST_{\tau, m}$ networks can train consistently to high accuracy with activation sparsity of 90\%.
> >
> > ___
> >
> > **Comment:**
> >
> > "The authors should also discuss the effect of adding residual connections on the stability of the network training. As residual-networks has been the main solution for training very deep networks, the authors should clarify whether their results also apply for residual networks."
> >
> > **Response:**
> >
> > In this paper we develop present and analyse EOC theory for sparsifying activation functions for feedforward and convolutional networks only. It unfortunately does not directly apply to residual networks, which we agree were of course a very important architectural development for stably training deep nets. However, ResNets would need their own EOC theory worked out for these activation functions. This is an important and promising avenue for future work, and we have noted this as such in the conclusion. However, we feel that this falls beyond the scope of the present paper, and that the insights and contributions of the paper in its current form merit publication in their own right.

---

> > > ### Author Response · Authors · 2023-11-15
> > > **Authors' response part 3**
> > >
> > > **Comment:**
> > >
> > > "Additionally, it seems that some of the experiments were left out of the main paper, and only appear in the appendix (for example, studying the CST activation and comparing the clipped activations with non-clipped ones). These are key experiments in the paper and should appear in the main text."
> > >
> > > **Response:**
> > >
> > > Thank you for this suggestion. Putting $ReLU_\tau$, $ST_\tau$, and $CST_\tau$ experimental results in the Appendix was originally done purely due to space constraints, but we acknowledge and agree that it is important for all key experimental results to appear in the main paper. In order to make space, we simplified and combined Figures 2 and 3, and so we have now included all the important experimental results in Table 2.

---

> > > ### Comment · Reviewer_FeGe · 2023-11-21
> > >
> > > Thank you for your response.
> > >
> > > These experiments are indeed more convincing, showing the potential applicability of the suggested sparsity inducing activation function. Therefore, I have raised my score.
> > >
> > > About the comparison to ResNets: I understand that presenting EOC analysis for ResNets is beyond the scope of the paper, but I believe that the authors should add experiments comparing networks with and without residual connections, studying how the modification of the activation function interacts with the residual connections. It is interesting to understand whether the stability issue of sparse networks can be solved by adding residual connections.

---

### Official Review · Reviewer_qoAp · 2023-10-31

**Soundness:** 3 good
**Presentation:** 3 good
**Contribution:** 3 good
**Rating:** 6
**Confidence:** 3

**Summary:**

The authors study sparsity inducing non-linear activation functions in neural network training. They provide a theoretical analysis of the instability effects for using shifted ReLU and SoftThreshold when training FeedForward networks and CNNs, motivating their clipped versions of these functions. The authors use the previously introduced EoC initialization and show why training is stable, and becomes unstable with the introduction of a modified ReLU activation function that should induce sparsity.
To remedy the instability in training a further modification (clipping) is introduced and proven to be theoretically stable. They then demonstrate the feasibility of their CReLU and CST activation functions for training deep and wide networks on MNIST and CIFAR10, showing only minor degradation in prediction accuracy with tunable levels of activation sparsity.

Overall, the paper is nicely written and relatively easy to follow, despite the math-heavy theoretical section. The argumentation on instability of the shifter ReLU and SoftThreshold seems valid, and the experiments, though not extensive, provide a proof-of-concept of the author’s claim. The idea of clipping the activation functions to achieve stability is as simple as it is effective, providing a valid contribution that can be actually translated into application.  However, a bit more in-depth evaluation of certain aspects would be nice. The findings are interesting, though presentation is lacking, as well as more exploration of the introduced concept.

**Strengths:**

- The proposed clipped functions are an easy and natural extension of already existing activation functions.
- The introduced parameters tau and m are able to control sparsity and training stability as shown theoretically and experimentally.

**Weaknesses:**

- The stddev in table 2 are a bit odd to me, either a higher numerical precision is needed, or it needs to be explained why the are many cases of zero std.
- Results on ST are only shown in the appendix. They should be shown and discussed in the main paper, given that they are an essential part in the rest of the manuscript.
- the higher sparsity regime in table 2 (s=0.85) needs to be explored/explained more, there are interesting things happening.
- There is no comparison to existing methods, but the authors clearly describe the lack of related research.
- No source Code provided, implementation details on experiments only in the appendix

**Questions:**

- I don’t fully understand the meaning of Figure 4 and it is not sufficiently discussed in the manuscript.
- The important EoC concept is not motived and explained enough.

---

> ### Author Response · Authors · 2023-11-15
> **Authors' response**
>
> We are very grateful to the reviewer for their careful reading of our paper, thoughtful feedback, and constuctive suggestions. Below we respond to the concerns and/or questions raised.
>
> ___
>
> **Comment:**
>
> "The stddev in table 2 are a bit odd to me, either a higher numerical precision is needed, or it needs to be explained why the are many cases of zero std."
>
> **Response:**
>
> Indeed most of the 0.00 standard deviations were simply because the results in the table were rounded to 2 decimal places. We have amended Table 2 to include higher precision to make the results clearer. 0.00 standard deviation was only accurate to higher precision as well in the cases when all of the seeds completely fail to train at all. Thank you for this suggestion.
>
> ___
>
> **Comment:**
>
> "Results on ST are only shown in the appendix. They should be shown and discussed in the main paper, given that they are an essential part in the rest of the manuscript."
>
> **Response:**
>
> Thank you for this suggestion. Putting $ReLU_\tau$, $ST_\tau$, and $CST_{\tau, m}$ experimental results in the Appendix was originally done purely due to space constraints, but we acknowledge and agree that it is important for all key experimental results to appear in the main paper. In order to make space, we simplified and combined Figures 2 and 3, and so we have now included all the important experimental results in Table 2.
>
> ___
>
> **Comment:**
>
> "The higher sparsity regime in table 2 (s=0.85) needs to be explored/explained more, there are interesting things happening."
>
> **Response:**
>
> We agree that the high sparsity regime is very interesting! But we suggest that our analysis of the impact of $s$ and $m$ on the shape of the corresponding variance maps and the related failure modes accounts for the results quite nicely. In particular, we see that initially increasing $m$ is necessary to achieve good accuracy at high sparsity, as expected. However, as predicted by the theory, increasing $m$ too much results in $V'(q^*)$ and $V''(q^*)$ together being too large, causing $q^*$ to fail to be stable in practice. This relates to your other question about Figure 4. The phenomenon described here is exactly what is illustrated in Figure 4, which shows the Variance maps of $CReLU_{\tau,m}$ and $CST_{\tau, m}$ in the high sparsity, large $m$ regime. When the variance map curves of to trace or even cross the $q=V(q)$ line, the result is that $q^l$ does not converge to $q^*$, and instead remains larger than $q^*$ layer on layer. Figure 4 shows that the corresponding $\chi_1$ value in this case is larger than one, which causes exploding gradients and training failure. We have tweaked the wording at the bottom of page 8 in Section 4 to make it clearer which point from the main text Figure 4 is illustrating.
>
> If you still think something is not clear, could you perhaps explain what analysis you think is missing?
>
> ___
>
> **Comment:**
>
> No source Code provided, implementation details on experiments only in the appendix.
>
> **Response:**
>
> The implementation details were left to the appendix due to space constraints. We ask that this please not be considered a weakness of our paper.  We are working to make the source code available and will alert you if this is possible before the discussion period concludes.

---

### Official Review · Reviewer_auBc · 2023-11-01

**Soundness:** 4 excellent
**Presentation:** 4 excellent
**Contribution:** 4 excellent
**Rating:** 8
**Confidence:** 3

**Summary:**

The paper aims to encourage high sparsity in the activations of neural networks with the motivation to reduce computational cost. To this end, the authors study the activation dynamics induced by common sparsity inducing nonlinearities such as ReLU and Soft-Thresholding (ST) under random initialization. The authors, via the large width Gaussian process limit of neural networks, discover a training instability for ReLU and ST nonlinearities. They show that the instability can be resolved by clipping the outputs of these nonlinearities. The authors validate the theory through experiments and show that the modification allows training MLPs and CNNs with very sparse activations with no or little reduce in test accuracy.

**Strengths:**

* The writing is very clear and easy to follow.
* The theory is elegant.
* The theory works in practice and the authors effectively demonstrate being able to train neural networks while maintaining high activation sparsity.

**Weaknesses:**

Minor weaknesses:
* There could have been a study of the computational efficiency since that is the main motivation of the work.

**Questions:**

Questions:
- Is this the first time that the variance map equations are being derived for these non-linearities?

Minor:
- Plots of Figure 2 are missing axis labels and the plot legends are not readable on printed paper.

---

> ### Author Response · Authors · 2023-11-15
> **Authors' response**
>
> We are very grateful to the reviewer for their careful reading of our paper, thoughtful and positive feedback, and constructive suggestions. Below we respond to the concerns and/or questions raised.
>
> ___
>
> **Comment:**
>
> "Is this the first time that the variance map equations are being derived for these non-linearities?"
>
> **Response:**
>
> Yes, to the best of our knowledge this is the first time they have been derived.
>
> ___
>
> **Comment:**
>
> "Plots of Figure 2 are missing axis labels and the plot legends are not readable on printed paper."
>
> **Response:**
>
> Thank you for highlighting this. We have added axis labels and enlarged the legend font.
>
> ___
>
> **Comment:**
>
> "There could have been a study of the computational efficiency since that is the main motivation of the work."
>
> **Response:**
>
> Unfortunately it was not possible for us to perform an empirical study of the computational efficiency gains due to sparser activations, because the sparse operations necessary to leverage sparse activations are not yet well supported on the accelerator hardware on which we are running our experiments. However, better support  for efficient sparse operations is a high priority and ongoing research area for both deep learning hardware and software developers, given the potential performance gains. The  number of flops in each matrix-vector multiplication $AB$ for $A \in \mathbb{R}^{m \times n}$, $B \in \mathbb{R}^{n \times d}$  in the forward pass could in theory shrink from $\mathcal{O}(mnd)$ to $\mathcal{O}(msd)$.

---

> > ### Comment · Reviewer_auBc · 2023-11-20
> >
> > Thank you for the response. I very much enjoyed reading your paper and I retain my positive opinion of the work.

---

### Official Review · Reviewer_TyEd · 2023-11-02

**Soundness:** 3 good
**Presentation:** 2 fair
**Contribution:** 3 good
**Rating:** 6
**Confidence:** 2

**Summary:**

The paper studies the effect of sparsity on the activation function for deep neural network initialization using the existing dynamical isometry and edge of stability theory. In particular, the authors compute the so-called variance map and correlation maps for sparse activating functions, namely, shifted ReLU and soft thresholding, and interpret the shape of these maps, in particular, the values $V'(q^*)$ and $V''(q^*)$ to explain the failure. Then they propose magnitude clipping as a remedy and empirically show that with these magnitude-clipped sparse activation functions, it is possible to train the deep net without losing test accuracy and with high test fractional sparsity.

**Strengths:**

The paper introduces two natural classes of activation functions with a tunable parameter $\tau$. I think understanding what activation function works better for which purpose is an active and fascinating area which I also believe is to appeal to general interest. Sparsity is particularly an important goal to achieve for modern large deep learning. Making sure that the network has non-exploding or non-vanishing gradients at initialization is indeed a sufficient condition for the applicability of an activation function.

The introduction is well-written.

On the flip side, I am not entirely sure why sparse ReLU fails to train. Table 2 only shows results for magnitude clipping for CReLU and the usual ReLU with $\tau=0$.

**Weaknesses:**

I have three major questions that I wasn't able to resolve by reading the main text only.

1. What is the criterion on $V_\phi$ for successful training? For example, let's see Figure 2. Which of the shapes are good and expected to train well vs which are the ones that are expected to fail? I see that for $\tau=1$ the curves have higher curvature. In particular, the blue curve intersects the line $x=y$ at one point where the derivative is non-zero but the curvature is positive. Is this expected to fail because the derivative is non-zero? I found the explanations in the text somehow repetitive and hard to parse. Can the authors explain the criterion on $V_\phi$ in words just from Figure 2?

2. Can the authors please provide experiments with CReLU $m=0$ as well and also for ReLU in Table 2? Why is there only one row for ReLU?
As the table stands now, I am not convinced that sparse activation functions without magnitude clipping fail to train. Is the 'unstable training dynamics' reported in the paper for very large $s$ and small $m$ as claimed in the conclusion?

**Questions:**

Also, I do not understand the heuristics given in Section 3.1 for how to choose $m$. I understand the dependence of $V'(q^*)$ and $V''(q^*)$ on the magnitude value $m$ is non-trivial from Figure 4. Still, the curves follow regular shapes so maybe it is possible to give simple heuristics based on Figure 4?

I will consider increasing my score based on the author's response to my questions.

------------------------------------------------------------------------------------------------

Post-rebuttal: The authors sufficiently addressed my concerns. I am leaning acceptance.

---

> ### Author Response · Authors · 2023-11-15
> **Authors' response part 1**
>
> We are very grateful to the reviewer for their careful reading of our paper, thoughtful feedback, and constructive suggestions. Below we respond to the concerns and/or questions raised.
>
> ___
>
> **Comment:**
>
> "I am not entirely sure why sparse ReLU fails to train. Table 2 only shows results for magnitude clipping for CReLU and the usual ReLU with $\tau=0$." and "What is the criterion on $V_\phi$ for successful training? For example, let's see Figure 2. Which of the shapes are good and expected to train well vs which are the ones that are expected to fail? I see that for $\tau=1$ the curves have higher curvature. In particular, the blue curve intersects the line x = y at one point where the derivative is non-zero but the curvature is positive. Is this expected to fail because the derivative is non-zero? I found the explanations in the text somehow repetitive and hard to parse. Can the authors explain the criterion on $V_\phi$ in words just from Figure 2?"
>
> **Response:**
>
> According to the theory, two things are required for stable training:
>
> 1. we require that $\chi_1 = 1$, which guards against vanishing and exploding gradients as well as instability to input perturbations at initialisation.
>
> 2. we require that the variance map $V(q)$ is sufficiently stable around $q^*$ such that $q^l$ does in practice converge to and remain at $q^*$. This is a necessary requirement for the EOC initialisation in practice since $q^*$ is used in the calculation of the values of $\sigma_w$ and $\sigma_b$ in order to ensure $\chi_1 = 1$. If in practice $q^l$ does not stably converge to  $q^*$, and instead grows larger, then we no longer have $\chi_1=1$ and experience the associated training failure modes of exploding or vanishing gradients.
>
> The problem we have shown with $ReLU_\tau$ and $ST_\tau$ is precisely that it is impossible for them to satisfy both criteria $\chi_1=1$ and $q^l$ converging stably to $q^*$.
>
> In light of the above, we agree that the original Figures 2 and 3 could be misleading, since they included variance maps for $ReLU_\tau$ and $ST_\tau$ which did not correspond to EOC initialisation. To address your question and to improve clarity on this point we have combined Figures 2 and 3 into a single figure, now showing only the variance maps for both activation functions corresponding to $\chi_1=1$. We have also edited Section 2 of the paper to make this point more clearly (in particular see paragraphs 2, 3, 4, and 5 of Section 2 in the updated version). We think it reads better now, so thank you for this helpful feedback.
>
> ___
>
> **Comment:**
>
> "Why is there only one row for ReLU?"
>
> **Response:**
>
> Standard ReLU was simply included in Table 2 as a baseline, a commonly used activation function against which we can compare the accuracy and activation sparsity in our experiments.

---

> > ### Author Response · Authors · 2023-11-15
> > **Authors' response part 2**
> >
> > **Comment:**
> >
> > "Can the authors please provide experiments with CReLU m = 0 as well and also for ReLU in Table 2? ... As the table stands now, I am not convinced that sparse activation functions without magnitude clipping fail to train."
> >
> > **Response:**
> >
> > Indeed Table 2 is not intended to provide evidence that $ReLU_\tau$ and $ST_\tau$ fail to train at high sparsities. Due to space constraints, the experiments showing that $ReLU_\tau$ and $ST_\tau$ networks do indeed fail to train at higher sparsities were included only in the appendix in Table 3, top of page 17, in the original version. We fully agree that the pointer to these experimental results was not sufficiently clear (it was made on page 4 at the end of the following sentence: ``To make matters worse, ...  and consequently prove effectively impossible to train for large sparsity; see App. C."). Having made some additional space available by merging Figures 2 and 3 as described above, we have now included *all* experimental results in Table 2. For ease of reference, these are the experimental results for $ReLU_\tau$ and $ST_\tau$:
> >
> > |            | $s$  | $\tau$ | $m$ | $V'_\phi$ at q* | $V''_\phi$ at q*  | DNN accuracy mean | DNN accuracy std   | DNN sparsity mean | DNN sparsity std   |   CNN accuracy   |  CNN sparsity     |
> > |------------|------|--------|-----|----------------|-----------------|------|-------|------|-------|------|------|
> > | $ReLU_\tau$ | 0.50 | 0.00   | N/A | 1.0            | 0.0             | 0.94 | 0.002 | 0.50 | 0.001 | 0.70 | 0.52 |
> > |            | 0.60 | 0.25   | N/A | 1.0            | 0.12            | 0.76 | 0.37  | 0.49 | 0.27  | 0.68 | 0.6  |
> > |            | 0.70 | 0.52   | N/A | 1.0            | 0.3             | 0.10 | 0.00  | 0.00 | 0.00  | 0.1  | 0.0  |
> > | $ST_\tau$      | 0.5  | 0.67   | N/A | 1.0            | 0.43            | 0.10 | 0.00  | 0.00 | 0.00  | 0.1  | 0.0  |
> > |            | 0.6  | 0.84   | N/A | 1.0            | 0.59            | 0.10 | 0.00  | 0.00 | 0.00  | 0.1  | 0.0  |
> > |            | 0.7  | 1.04   | N/A | 1.0            | 0.81            | 0.10 | 0.00  | 0.00 | 0.00  | 0.1  | 0.0  |
> >
> >
> >
> > The results clearly show the failure of the $ReLU_\tau$ networks to train once sparsity reaches 70\%, and the failure of $ST_\tau$ at even 50\% sparsity.
> >
> > While $CReLU_{\tau,m}$ with $m=0$ will be just the 0 function for any $\tau$, and so would not work as a candidate activation function; potentially we are misunderstanding the request "experiments with CReLU m = 0."
> >
> > ___
> >
> > **Comment:**
> >
> > "Is the `unstable training dynamics' reported in the paper for very large s and small m as claimed in the conclusion? Is the 'unstable training dynamics' reported in the paper for very large s and small m as claimed in the conclusion?"
> >
> > **Response:**
> >
> > Table 2 shows the failure to consistently train when combining the largest $s$ together with smallest $m$ considered in the experiments.  Table 2 also shows that stability is recovered by decreasing $m$ and/or $s$ sufficiently.
> >
> > ___
> >
> > **Comment:**
> >
> > "Also, I do not understand the heuristics given in Section 3.1 for how to choose $m$. I understand the dependence of $V'(q^*)$ and $V''(q^*)$ on the magnitude value m is non-trivial from Figure 4. Still, the curves follow regular shapes so maybe it is possible to give simple heuristics based on Figure 4?"
> >
> > **Response:**
> >
> > As you say, the dependence of the shape of $V_\phi$ on $s$ and $m$ is non-trivial, and we are hesitant to suggest a simple and clean heuristic, since as far as we know none can be derived analytically. Our experiments in Table to 2 suggest that a decent starting point for fully connected networks would be to maximise $m$ for a given sparsity subject to ensuring that $V''(q^*) \lessapprox 0.2$, while allowing $V''(q^*) \lessapprox 0.4$ appears to work in the CNN case.
> >
> > ___
> >
> > **Comment:**
> >
> > "I will consider increasing my score based on the author's response to my questions."
> >
> > **Response:**
> >
> > Thank you very much. We hope that we have sufficiently addressed your questions and suggestions.

---

### Author Response · Authors · 2023-11-20
**Follow-up on review responses**

Thank you to all reviewers once again for your valuable time and insightful comments which have helped us improve the paper further.

As the deadline for the Author/Reviewer discussion is approaching, we would very much appreciate it if you could let us know whether you are satisfied with our answers to your questions and the corresponding updates we have made to the paper, reflected in the most recent revision. If so, we would humbly ask that you consider adjusting your scores as you feel appropriate. We are of course happy to provide any additional clarifications that you may need.

---

### Meta-Review · Area_Chair_ZEW7 · 2023-12-04

**Metareview:**

The paper uses ideas from the infinite-width limit and Gaussian process to propose stable alternatives for sparsity-inducing activations. The overall response by the reviewers is positive. The results are novel, and the empirical analysis for justifying the ideas has been performed thoroughly. There were a few concerns, such as the criterion on $V_\phi$ for successful training and the heuristic for choosing $m$, which were sufficiently addressed by the authors.

**Justification For Why Not Higher Score:**

The empirical analysis is done meticulously, justifying each step in the construction. Some theoretical results are novel but need to be stronger for a higher score.

**Justification For Why Not Lower Score:**

The paper is good and the reviewers all agree on acceptance.

---

### Decision · Program_Chairs · 2024-01-16

Accept (poster)